# A BRC1-modulated switch in auxin efflux accounts for the competition between Arabidopsis axillary buds

Zoe Nahas[¤a], Anthony John Bridgen, Torkel E. Loman[¤b], Jean Dillon,
Katie Abley, Dora L. Cano-Ramirez, Fabrizio Ticchiarelli, Madeleine Seale,
James C. W. Locke*, Ottoline Leyser*

Sainsbury Laboratory, University of Cambridge, Cambridge, United Kingdom

¤a Current address: Centre for Integrative Genomics, Faculty of Biology and Medicine, University of Lausanne, Lausanne, Switzerland
¤b Current address: Mathematical Institute, University of Oxford, Oxford, United Kingdom
* ol235@cam.ac.uk (OL); james.locke@slcu.cam.ac.uk (JCWL)

## Abstract

As part of their modular development, plants continuously adapt their shoot branching architecture according to environmental conditions. This occurs by regulating the activity of axillary buds established in each leaf axil. Whether a bud grows into a shoot depends partly on the presence of other active shoots, which can inhibit bud activation. This systemic coordination is proposed to be mediated by the transport network of the plant hormone auxin, with buds competing to establish sustained transport of auxin, termed canalized auxin transport, into the main stem. A second hormone, strigolactone, tunes this competition by influencing the removal of the PIN1 auxin export protein from the plasma membrane, and hence the dynamics of canalization. Strigolactone also regulates the expression of another key regulatory hub, the bud-expressed transcription factor BRANCHED1 (BRC1). The interplay between auxin transport and BRC1 in regulating bud activity is poorly understood. Here, we investigate this interplay in the context of competition between buds, using Arabidopsis explants with two axillary buds as a minimal system. Using experimental data, we develop a mathematical model of bud-bud competition in which BRC1 influences the establishment of canalized auxin transport by regulating the basal rate of auxin efflux in buds. We identify single model parameters that plausibly correspond to the dual impact of strigolactone on *BRC1* expression and PIN1. We show that modulating these two parameters reproduces the dynamics of bud growth and bud-bud competition observed in relevant mutants and treatments. Our model produces testable hypotheses, which we validate by generating a chimeric PIN1 auxin transporter with impaired strigolactone sensitivity, helping us uncouple the effects of strigolactone on PIN1 and *BRC1*. These results support the hypothesis that BRC1 influences local bud competitiveness by downregulating the basal rate of auxin efflux in buds.

**Data availability statement:** All data and code are available at https://doi.org/10.17863/CAM.120831.

**Funding:** ZN, AB, FT, JD, MS, OL were supported by the Gatsby Charitable Foundation No. GAT3272C awarded to OL. KA, DCB, TL, JL were supported by Gatsby Charitable Foundation (https://www.gatsby.org.uk/plant-science) No. GAT3395/PR5 awarded to JL, with the work in JL lab also supported by the Biotechnology and Biological Sciences Research Council (https://www.ukri.org/councils/bbsrc/) grant awards no. BB/V006088/1 and no. BB/X020126/1. The funders had no role in study design, data collection and analysis, decision to publish, or preparation of the manuscript.

**Competing interests:** The authors have declared that no competing interests exist.

**Abbreviations:** ABA, abscisic acid; CDS, chimeric coding sequence; PATS, polar auxin transport stream; RGI, relative growth index; SLIC, sequence and ligation-independent cloning.

Together with the systemic feedbacks in the auxin transport network, this enables plants to adjust dynamically the number and location of growing branches.

---

## 1. Introduction

Plants achieve extraordinary flexibility of form through their modular growth, allowing their body plan to adapt to the prevailing environmental conditions. Above ground, plants continuously tune their shoot architecture by regulating the activity of axillary buds, which are derived from meristems established in the axil of each leaf [1]. Whether an axillary bud stays dormant or grows out into a shoot depends on the integration of external cues, such as nutrient availability and light quality, and internal cues, such as the activity of other shoot apices on the plant [2,3].

Two hubs have been identified that integrate bud regulatory signals. The first hub is the regulated expression of BRANCHED1 (BRC1), a bud-expressed transcription factor that acts as a local inhibitor of bud activation, as evidenced by its highly branched mutant phenotype [4,5]. Among its many regulators [6,7], BRC1 expression is up-regulated by the plant hormone strigolactone (Fig 1A) [4,9–11]. Strigolactone signals via its receptor D14, triggering the degradation of a small family of proteins, which in Arabidopsis are named SUPPRESSOR OF MAX2 (SMXL) 6, 7, and 8 (hereafter SMXL678) [12–15]. SMXL6 represses BRC1 transcription directly [16].

Downstream of BRC1, several direct targets have been identified, including the MAX1 gene, which encodes a strigolactone biosynthesis enzyme [17,18], and several genes associated with the growth-inhibiting hormone abscisic acid (ABA) [18,19–21]. Although BRC1 expression correlates strongly with bud inhibition, it is neither necessary nor sufficient for bud inhibition. In Arabidopsis, two paralogues, BRC1 and BRC2, carry out the function, with BRC1 playing a more significant role in bud inhibition. In the brc1brc2 double mutant, whilst many buds activate, some remain dormant. Moreover, buds expressing high levels of BRC1 can exhibit wild-type outgrowth [22]. Hence, rather than straightforwardly inhibiting bud activation, BRC1 may affect the bud activation potential [22].

The second hub for signal integration is the auxin transport network. Unlike local BRC1-based regulation, the auxin transport network acts systemically, enabling the coordinated activity of shoot apices across the plant [23–25]. Auxin is largely synthesized in the young expanding leaves of active shoot apices [26], and is transported basipetally in the polar auxin transport stream (PATS). PATS activity is mediated by basally localized auxin transporters of the PIN-FORMED (PIN) family, particularly PIN1, primarily associated with stem vascular tissues [27–29].

Auxin export from the axillary buds into the main stem correlates with, and is thought to be necessary for, sustained bud activity [25,30–35]. The establishment of auxin export occurs early in bud activation, likely through the process of auxin transport canalization [33,36–38]. Auxin transport canalization is hypothesized to be driven by an autocatalytic feedback between auxin flux and the polar accumulation of PIN1 auxin export proteins at the plasma membrane in the direction of the flux, promoting increased and polarized auxin flux [23,39–42]. Through this mechanism, auxin

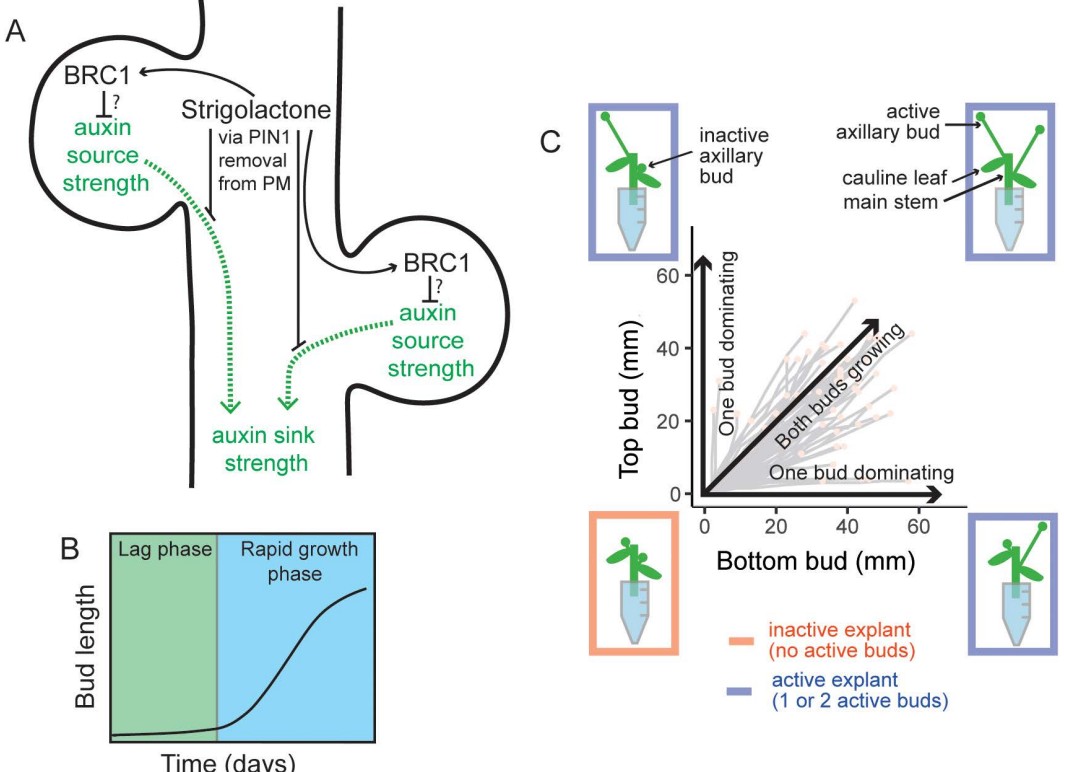

**Fig 1. 2-node explants capture key properties of bud regulation. (A)** A stem segment with two axillary buds illustrates two regulatory hubs controlling shoot branching (i) local expression of the transcription factor BRC1, a repressor of bud activation, and (ii) systemic regulation of the auxin transport network. A canalization-based model of shoot branching postulates that bud activation requires the establishment of canalized auxin transport from the bud into the main stem, the dynamics of which is influenced by autocatalytic feedback in auxin flow between the bud and the stem, and the relative auxin source and sink strengths of the bud and stem, respectively. The relationship between BRC1- and auxin-transport-mediated regulation is not known. **(B)** Arabidopsis bud activation occurs in at least two phases: a slow-growing lag phase, then a switch to rapid outgrowth. Typical timescale 10–12 days. Modified from Nahas and colleagues [8]. **(C)** Diagram illustrating the four possible growth outcomes for bud activation on 2-node explants, and their representation in a Mitchison plot. Mitchison plots present the length of the top bud versus that of the bottom bud over time in each explant. Explants where at least one bud grows are termed active, otherwise, they are inactive. Within active explants, there are three possible outcomes: both buds grow, or only either the top or the bottom bud grows. All graphics were drawn by hand using Adobe Illustrator.

moving passively from a source to a sink, through an initially broad field of cells, promotes the formation of progressively narrower files of cells with highly polar auxin transport [43,44].

In a canalization-based model of bud regulation, buds are auxin sources and can establish canalized auxin transport from the bud into the stem, and hence activate, only if the stem is a strong enough sink for auxin [25]. This happens for example, when the primary shoot apex, which exports auxin into the main stem, is removed, and hence auxin is depleted from the main stem. Conversely, high auxin in the main stem, coming from the primary apex and/or growing axillary shoots, makes the stem a poor sink for auxin, reducing auxin flux out of the bud, thereby preventing the establishment of canalized auxin transport from the bud into the stem [25,30,33,34,45]. Since auxin biosynthesis is under negative feedback control, failure to export auxin reduces auxin synthesis in the bud, capping bud auxin source strength [26].

Collectively, these properties of auxin transport generate a competitive bistable switch in bud activation [25]. Once past a tipping point, auxin transport canalization is hard to reverse. Consistent with this, only small buds can be inhibited by high auxin in the main stem [46]. At the level of bud growth dynamics, bud activation follows a slow-growing lag phase followed by a switch to rapid outgrowth (Fig 1B). Buds lose sensitivity to apical auxin at the same time as they switch from

the slow-growing lag phase to rapid outgrowth [8], suggesting that the lag phase corresponds to the time during which buds establish canalized auxin transport into the stem.

The competitive nature of the bud activation switch is clearly visible in 2-node explants, such as Arabidopsis stem segments with only two nodes and their associated axillary buds (Fig 1C). These exhibit four alternative bud activity outcomes: (i) neither bud grows, (ii) only the top bud grows, (iii) only the bottom bud grows, (iv) both buds grow out to produce branches [45], similar to classical 2-branch models [32,36,45,47,48]. The outcomes can be visualized by plotting the length of the top bud against the length of the bottom bud (so-called Mitchison plots) (Fig 1C). Where only a single bud grows, it quickly establishes dominance over the other bud, preventing its growth. Within a canalization-based model, if one bud rapidly establishes auxin export, it can prevent the establishment of canalized auxin transport from the other bud, but if both buds establish canalized auxin transport near-simultaneously, then neither will be able to inhibit the other [25].

Further evidence supporting the auxin transport canalization model of bud regulation comes from analyzing the effect of strigolactone on PIN auxin transporters. Strigolactone, in addition to up-regulating *BRC1* expression, triggers the removal of PIN1 auxin transporters from the plasma membrane through an unknown transcription-independent mechanism (Fig 1A) [24,49]. Given that the dynamics of the establishment of canalized auxin export are influenced by the relative rates of insertion and removal of PIN1 from the plasma membrane, the strigolactone-mediated removal of PIN1 could inhibit bud activity by making canalization harder to achieve. Consistent with this idea, strigolactone inhibits the formation of the narrow PIN1-expressing files of cells typical of canalized tissue [50]. A computational model in which strigolactone increases PIN1 removal from the plasma membrane captures the highly branched phenotypes of strigolactone deficient mutants [25]. The model also explains non-intuitive branching phenomena, including the ability of strigolactone to promote or inhibit shoot branching, depending on the auxin transport status of the plant [49]. Moreover, this mode of action accounts for the observation that strigolactone supply to 2-node explants results predominantly in a single active branch, rather than inhibition of both branches [51].

Although it has been shown that *BRC1* and the auxin transport network act as hubs for signal integration in shoot branching control, with both being regulated by a wide range of factors, including strigolactone, it is poorly understood how they simultaneously regulate bud activity. We recently demonstrated that in isolated 1-node explants, *BRC1* influences the length of the lag phase during bud activation, suggesting that it may affect the rate of establishment of auxin transport canalization out of the bud [8]. Therefore, we developed a unified model incorporating this hypothesis and assessed whether it could capture the dynamics of growth and bud-bud competition in 2-node explants. We find that our computational model captures multiple bud behaviors across a wide range of genotypes and treatments, and generates testable predictions, which we largely validate. Taken together, this work integrates the *BRC1* and auxin transport regulatory hubs into a single model, combining systemic and local modes of shoot branching regulation.

## 2. Results

### 2.1. Strigolactone-mediated *BRC1* expression inhibits bud activation in 1-node explants

The dual action of strigolactone on *BRC1* and PIN1 provides an opportunity to assess the relationship between these two modes of bud regulation. Strigolactone signaling leads to the degradation of SMXL678 proteins [12,13,15], such that the *smxl678* triple mutant has constitutively active strigolactone signaling, with high levels of *BRC1* expression and PIN1 removal from the plasma membrane [14] (Fig 2A).

To test the impact of *BRC1* on bud outgrowth dynamics, we compared bud behaviors on 1-node explants of Col-0, *brc1brc2*, *smxl678*, and *brc1brc2 smxl678*. We use the *brc1brc2* double mutant throughout our work because, although a substantial body of evidence suggests that BRC1 plays a dominant role, there is likely some redundancy between BRC1 and BRC2 [4,5,19]. Throughout, we refer to the effect of BRC1, but this may include an effect of BRC2. Buds were defined as active if their growth rate reached at least 2.5 mm/day. Unlike on Col-0 and *brc1brc2* explants, *smxl678* buds did not activate within the timeframe of our experiment (Fig 2B). However, their activation was fully restored in the *brc-1brc2smxl678* quintuple mutant, suggesting that the lack of growth of *smxl678* buds in 1-node assays is due primarily to

 

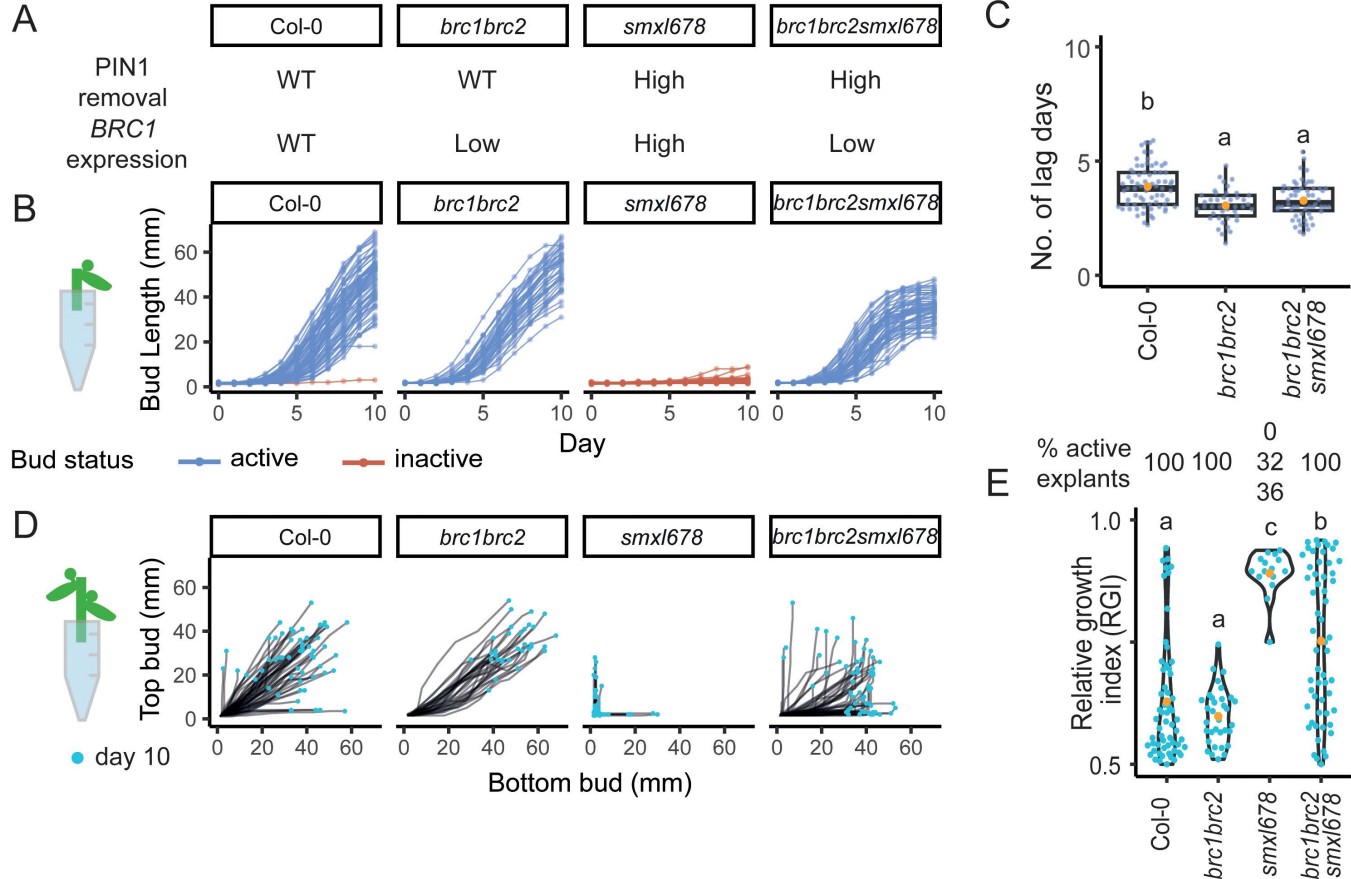

**Fig 2. The inhibition of *smxl678* buds is rescued by *brc1brc2* in 1-node but not in 2-node explants. (A)** PIN1 removal and *BRC1* expression phenotype in Col-0, *brc1brc2*, *smxl678*, and *brc1brc2smxl678*. **(B)** Bud length on 1-node explants measured daily for 10 days. Active buds (blue) are defined as those with a maximum growth rate > 2.5 mm/day, n = 41–72. **(C)** Number of lag days (number of days before growth rate > 2.5 mm/day) for active buds, n = 41–71 **(D)** Mitchison plots of top vs. bottom bud lengths on 2-node explants, measured daily for 10 days. The final length of buds is shown with a blue dot, n = 33–67. **(E)** Percentage of active explants (explants with at least one active bud) from three experimental repeats (where only one percentage is indicated, it is the same across the repeats). Below, relative growth index (RGI) of active explants at day 10. The RGI is defined as the length of the longest bud divided by the total length of both buds. The orange dot indicates the mean, n = 15–58. Letters represent statistically significant differences for *p* < 0.05 from a multi-level model of three experimental replicates with pairwise comparisons and Bonferroni correction. Data and analysis underlying this figure can be found at https://doi.org/10.17863/CAM.120831.

high levels of *BRC1* expression. We calculated the duration of the initial slow-growing lag phase, defined as the number of days before bud growth rate exceeds 2.5 mm/day (Fig 2C). As previously reported, *brc1brc2* mutants have a shorter lag phase than Col-0 [8], which was also the case for *brc1brc2smxl678* (Fig 2C). Therefore with respect to early bud activation, *brc1brc2* is fully epistatic to *smxl678* in this assay. Interestingly, branches on *brc1brc2smxl678* explants consistently stopped growing earlier than Col-0 or *brc1brc2* branches (Fig 2B), suggesting that BRC1-independent strigolactone signaling influences sustained bud growth.

## 2.2. Strigolactone-mediated regulation of bud-bud competition in 2-node explants is due to *BRC1*-dependent and -independent effects

To explore the role of *BRC1* in mediating competition between buds, we assessed the behavior of buds on 2-node explants of Col-0, *brc1brc2*, *smxl678*, and *brc1brc2smxl678*. The growth of buds on each explant is shown on Mitchison

plots (introduced in Fig 1C) (Fig 2D). We use the relative growth index (RGI) of active explants (defined as explants with at least one active bud) to compare bud behavior between genotypes. The RGI is the length of the longest bud as a ratio of the total length of both buds. An RGI of 0.5 indicates both buds growing equally, while an RGI of 1 indicates one bud dominating the other.

Col-0 explants exhibited all three possible outcomes for active explants (introduced in Fig 1C), with either one or both buds activating in each explant (Fig 2D). In *brc1brc2* explants, both buds activated on all explants (Fig 2D), resulting in a lower mean RGI than for Col-0, although here this difference is not statistically significant (Fig 2E). This is consistent with previous results [22,52], and with the high branching of *brc1brc2* plants [4]. While *smxl678* buds did not activate in 1-node explants, interestingly one bud did activate in a substantial number of 2-node explants (Fig 2D), resulting in very high RGIs (Fig 2E).

Importantly, unlike for 1-node explants (Fig 2B), *brc1brc2* was not fully epistatic to *smxl678* in 2-node explants, but rather in *brc1brc2smxl678* explants, one bud often dominated the other, resulting in an RGI intermediate between that of *brc1brc2* and *smxl678* (Fig 2D and 2E). As in 1-node explants, *brc1brc2smxl678* active apices stopped growing early. This was often followed by activation of the other bud, with this growth pattern clearly visible on the Mitchison plots (Fig 2D), consistent with the ability of one bud to inhibit the other in this genotype. Taken together, these results indicate that the high RGI of *smxl678* is due to a combination of *BRC1*-dependent and -independent effects of strigolactone-mediated bud inhibition.

## 2.3. Incorporating *BRC1* into a canalization-based model of bud-bud competition

Bud-bud competition is hypothesized to be based on competitive auxin transport canalization into the stem [25,32,45,53], and we previously hypothesized that the lag phase of bud growth corresponds to the time during which canalized auxin transport is established [8]. Given this, the impact of *brc1brc2* on the length of the lag phase and on the growth outcomes of buds in 2-node explants suggests that *BRC1* might influence auxin efflux from buds during the establishment of auxin transport canalization. The auxin transport network, and hence the developmental processes it regulates such as shoot branching, are heavily feedback-driven, making their dynamics difficult to intuit [54,55]. Mathematical modelling is a powerful tool to test which interactions can account for observed phenomena [56–58]. To test whether the hypothesis that *BRC1* influences the dynamics of auxin transport canalization can account for the behaviors we observe, we built a mathematical model of canalization-based bud-bud competition in 2-node explants.

The model consists of two self-activating and mutually-inhibiting buds (Fig 3A). It preserves the most conceptually important aspects of the auxin transport canalization model for shoot branching from Prusinkiewicz and colleagues [25]. These include a hysteretic bistable switch in bud activation caused by flux-regulated polar PIN accumulation dictated by a Hill function, combined with linear PIN removal. We simplified the formulation of auxin flux by using a formulation for the ratio of auxin influx to efflux [59]. We also combine parameters referring to PINs, auxin concentration, and auxin diffusion, into one efflux term (Mathematical appendix). This was possible because we abstracted the stem section joining the two buds, and assumed that buds have an equal and constant auxin concentration over time, given evidence of homeostatic feedback on auxin synthesis [26]. We also abstracted the top and bottom bud positions, representing the explant as symmetrical. We obtain the following formulation for the efflux of auxin, *E* and *F*, out of the top and bottom bud, respectively (Fig 3B):

$$\frac{dE}{dt} = v0 + v\frac{(SE)^n}{(SE)^n + (D(E+F)+K)^n} - \mu E$$

$$\frac{dF}{dt} = v0 + v\frac{(SF)^n}{(SF)^n + (D(E+F)+K)^n} - \mu F$$

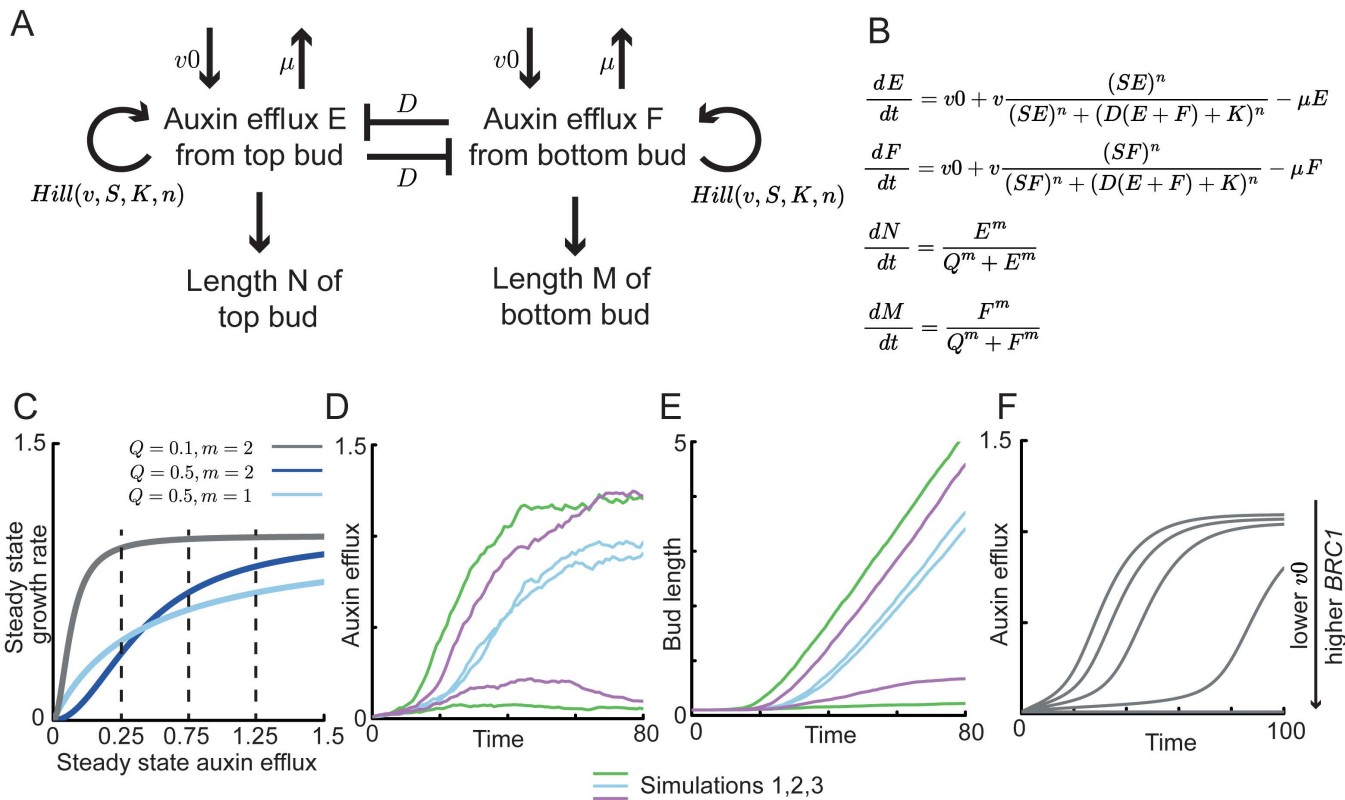

**Fig 3. Branching behaviors of 2-node explants can be captured by a model with self-activating and mutually-inhibiting buds. (A)** and **(B)** Model conceptualization and mathematical formulation. The interaction between two buds in a 2-node explant can be considered as a set of self-activating and mutually-inhibiting feedbacks on auxin efflux. Each bud promotes its own auxin efflux and inhibits efflux from the other bud. The auxin efflux $E$ and $F$ from the top and bottom bud, respectively, is influenced by three components (i) a basal rate of auxin efflux $v0$, (ii) a Hill function which creates a positive feedback on auxin efflux, where $v$ sets the maximum rate of auxin efflux, $S$ the strength of auxin efflux, $K$ the Hill saturation coefficient, $n$ the degree of non-linearity of the Hill function, $D$ the strength of the mutual inhibition between the auxin efflux of the two buds, and (iii) a linear decrease in auxin efflux, the strength of which is set by $\mu$. $E$ and $F$ influence bud lengths $N$ and $M$, respectively. The relationship between auxin efflux and growth rate is a Hill function, where $m$ influences the degree of nonlinearity, and $Q$ is the saturation coefficient. **(C)** Steady state growth rate as a function of the steady state of auxin efflux, for different values of $Q$ and $m$. Three grey vertical lines mark three steady states of auxin efflux at 0.25, 0.5, and 0.75. **(D)** and **(E)** Three stochastic simulations illustrating the model behaviors. Each set of simulations is represented with a different color. **(F)** Deterministic simulation showing the change in auxin efflux $E$ over time for 5 different values of $v0$: 0.01, 0.03, 0.05, 0.07, and 0.09. Scripts of simulations underlying this figure can be found at https://doi.org/10.17863/CAM.120831.

The change in auxin efflux is described by: (i) $v0$, a basal rate of efflux, which corresponds to non-polar auxin efflux of auxin out of the bud and depends on auxin levels, auxin transport activity, and auxin diffusion; (ii) a Hill function that represents the positive feedback of auxin efflux on itself, mirroring previous models of canalization, with $v$ as the maximum rate of efflux for the Hill function, $S$ the efficiency of feedback-driven auxin efflux, $n$ the Hill exponent that influences the degree of nonlinearity in the feedback, and $K$ a threshold parameter, and (iii) $\mu$, a linear decrease in efflux, which relates to both the removal of PINs from the plasma membrane and the degradation of auxin. The auxin efflux from one bud, conceptually contributing to the auxin concentration in the stem, dampens its own auxin efflux and that of the other bud at a strength that is proportional to $D$ and proportional to the sink strength of the main stem, which is determined by the sum of auxin efflux $E$ and $F$ from each bud. This captures the source/sink dynamics between the buds and the main stem.

To model bud growth rate, we postulate a relationship between growth and bud auxin efflux, captured by another Hill function:

$$\frac{dN}{dt} = \frac{E^m}{Q^m + E^m} \text{ and } \frac{dM}{dt} = \frac{F^m}{Q^m + F^m}$$

$N$ and $M$ are the length of the top and bottom bud, respectively. $Q$ is the Hill saturation coefficient and $m$ is the Hill exponent that influences the degree of non-linearity between auxin efflux and growth rate (Fig 3B). This relationship is informed by our previous analysis which suggested a positive relationship between auxin transport and growth rate in 1-node explants [8], though in some cases buds can grow rapidly with low steady state levels of auxin transport, for example in *brc1brc2smxl678* buds [22; Fig 2]. We therefore chose a parameter range where changes in steady state auxin efflux did not consistently cause a large change in growth rate (Fig 3C, dark blue).

To represent the initial dormancy of buds in 2-node explants, simulated buds were assigned equal very low starting values of auxin efflux. For simulations with relevant parameters, the positive feedback on auxin efflux increases the auxin efflux out of both buds. Without stochasticity, both buds switch to high auxin export in all simulations. However, if the positive feedback on auxin efflux exhibits some stochasticity, both buds initially increase their efflux, but one bud may gain a small advantage in its auxin efflux, which is amplified by the positive feedback, enabling one bud to establish canalized auxin export first, and to inhibit the canalization of the other bud (Fig 3D and 3E). Given the known correlation between auxin transport and sustained bud activity [25,30,31,33–35], and our previous hypothesis that the slow-growing lag phase corresponds to the time when buds establish canalized auxin transport [8], we chose parameters where the switch from slow to rapid growth coincides with an increase in auxin efflux (Fig 3D). This also entailed that inhibited buds, which export low auxin, exhibit little to no growth, consistent with experimental observations (Fig 3D and 3E) [45,8,60].

We next considered how to represent, in the model, the action of strigolactone on both PIN1 removal from the plasma membrane and *BRC1* expression. For the former, we postulated in a previous model that the parameter $\mu$ can be conceptualized as PIN1 removal (Fig 3A and 3B), such that an increase in $\mu$ models strigolactone-mediated PIN1 removal from the plasma membrane [25,49,51]. To represent *BRC1*, we considered that *BRC1* influences the length of the lag phase, which we hypothesize corresponds to the time of canalization establishment [8], and does not influence levels of bulk auxin transport through stem sections [61]. Given this, we assessed which parameters affect the timing of the switch to high auxin efflux during canalization, but not the steady state levels of auxin efflux. Both parameters $v0$ and $K$ meet these criteria (Figs 3F and S1). We chose to focus on $v0$, influenced by our previous observations that mutation in the *ABCB19* auxin efflux protein can partially suppress branching in *brc1brc2* mutants, and ABCB19 plausibly contributes to $v0$ [52]. In addition, BRC1 may affect auxin homeostasis by transcriptional regulation of *GH3* activity [18]. This could reduce auxin efflux in buds with high *BRC1* expression, making it harder for them to establish canalized auxin transport. Taken together, we hypothesize that the dual action of strigolactone via PIN1 removal and *BRC1* up-regulation can be represented by a simultaneous increase in $\mu$ and decrease in $v0$.

We identified a region of parameter space where the range of bud activation behaviors in the model match those in our experimental data, and where the dynamics of bud activation capture the basic properties described above. A range of ($v0$, $\mu$) slices deliver these core behaviors (S2A and S2B Fig). After systematically exploring parameter space (S2 and S3 Figs), we chose one ($v0$, $\mu$) slice for detailed analysis.

## 2.4. The model captures bud growth outcomes for a wide range of strigolactone-related genotypes and treatments

Strigolactone triggers PIN1 removal from the plasma membrane and promotes *BRC1* transcription, represented in the model by parameters $\mu$ and $v0$, respectively. To determine whether appropriate changes in $v0$ and $\mu$ were sufficient to capture the effects of relevant strigolactone mutations and treatments, we compared the behavior of buds on 2-node explants with cognate changes in $v0$ and/or $\mu$ in the selected ($v0$, $\mu$) slice, with all other parameters held constant.

Mitchison plots of bud measurements from experiments show that genotypes and treatments with variation in PIN1 removal and/or *BRC1* expression, exhibit variation in the relative proportion of explants where two versus only one bud activates (Fig 4A). In the strigolactone receptor mutant, *d14*, and in the *brc1brc2* mutant, both buds activate on all explants (Fig 4A, [52]). In Col-0 explants, either one or both buds grow (Fig 4A). Basal supply of the synthetic strigolactone rac-GR24 (hereafter GR24) or the *smxl678* background, a constitutive strigolactone signaling triple mutant, increases the level of competition between buds, increasing the frequency with which only one bud activates [51]. A qualitatively similar effect is seen with GR24 application and *smxl678* mutation in a *brc1brc2* mutant background, in which strigolactone is predicted to affect PIN1 removal from the plasma membrane and not *BRC1* expression (Fig 4A).

To test whether our experimental data in Fig 4A can be reproduced in the model via changes in *v0* and *μ* only, we mapped our experimental data onto the selected (*v0*, *μ*) slice of parameter space (Fig 4B (i)). Each pixel in the figure represents the modelled bud growth outcome at a given combination of *v0* and *μ* parameter values. Buds are described as active if they achieve canalized auxin export. The pixel color quantitatively represents the outcomes: (i) both buds growing (yellow), (ii) one bud growing (blue), and (iii) neither bud growing (red). The black lines describe the boundaries between areas of parameter space that have a different number of possible outcomes (i.e. a different number of stable steady states). In zone 1 both buds always grow. Zone 2 has two possible behaviors, with either one bud growing or both. The proportion of simulations with both buds growing versus one bud growing varies within zone 2. To reflect this shifting probability distribution, the pixel intensity is assigned as the ratio of outcomes from 50 stochastic simulations at each point in parameter space. The only behavior in zone 3 is one bud growing. In zone 4, either one bud grows, neither bud grows, or both buds grow, though for the starting values and timeframe of our simulations, the most common outcome is no buds growing, with occasionally one bud activating. The resulting (*v0*, *μ*) map of parameter space shows that *v0* and *μ* have similar effects but with opposite polarities, with increasing values of *μ* and reducing values of *v0* shifting the outcome from both buds growing, through to only one bud, and eventually fully inhibited explants.

We calculated the relative frequency of the various bud growth outcomes in the data. On the color map, there is a distribution of points, broadly across the tristable region (zone 2), where the frequency of bud growth outcomes in the data is the same as in the model. We plotted this distribution for each genotype/treatment (Fig 4B (ii)). Given we represent PIN1 removal as *μ* and hypothesize that *BRC1* expression can be represented as *v0*, we could constrain the placement of each genotype/treatment by considering their known phenotypes in terms of PIN1 accumulation and *BRC1* expression levels: (i) Under our hypothesis that *v0*, the basal rate of auxin efflux, is negatively regulated by *BRC1*, the *brc1brc2* mutant must have a higher value of *v0* than Col-0, and this high value will be common to all *brc1brc2* lines. (ii) PIN1:GFP membrane accumulation is the same in Col-0 and *brc1brc2* [11], and therefore their value of *μ* is likely also the same. (iii) Applying the synthetic strigolactone GR24 triggers the removal of PIN1 from the membrane and increases *BRC1* expression [4,10,11,49], so Col-0 + GR24 must have lower *v0* and higher *μ* than Col-0. (iv) Applying GR24 is assumed to increase *μ* by the same amount in Col-0 and *brc1brc2*, so Col-0 + GR24 and *brc1brc2* + GR24 are assumed to be on the same vertical line. (v) *smxl678* and *brc1brc2smxl678* have the same high PIN1 removal, such that they should be located on the same vertical line at a higher *μ* value than Col-0. (vi) Given its constitutive active strigolactone signaling, *smxl678* is modelled as more effective than GR24 treatment at increasing *μ* and decreasing *v0*.

These relationships create three vertical bars determining a value of *μ* for (i) Col-0 and *brc1brc2*, (ii) GR24 treatments, and (iii) *smxl678* backgrounds. In addition, three horizontal bars represent (i) the value of *v0* in *brc1brc2* mutant backgrounds, (ii) in Col-0, and (iii) after GR24 application, with *smxl678* plotted at an even lower *v0* value. Given these relationships, the values of *v0* and *μ* that best fit the pixel distribution are shown by the dark grey bars in Fig 4B (iii). The genotypes and treatments were placed at the intersection of these bars, each shown with a colored dot. The strigolactone receptor mutant, *d14*, has very low *BRC1* expression and low PIN1 removal [11], so was placed at the same low *v0* value as the *brc1brc2* mutants, with a lower *μ* value than Col-0 (Fig 4B (iii)).

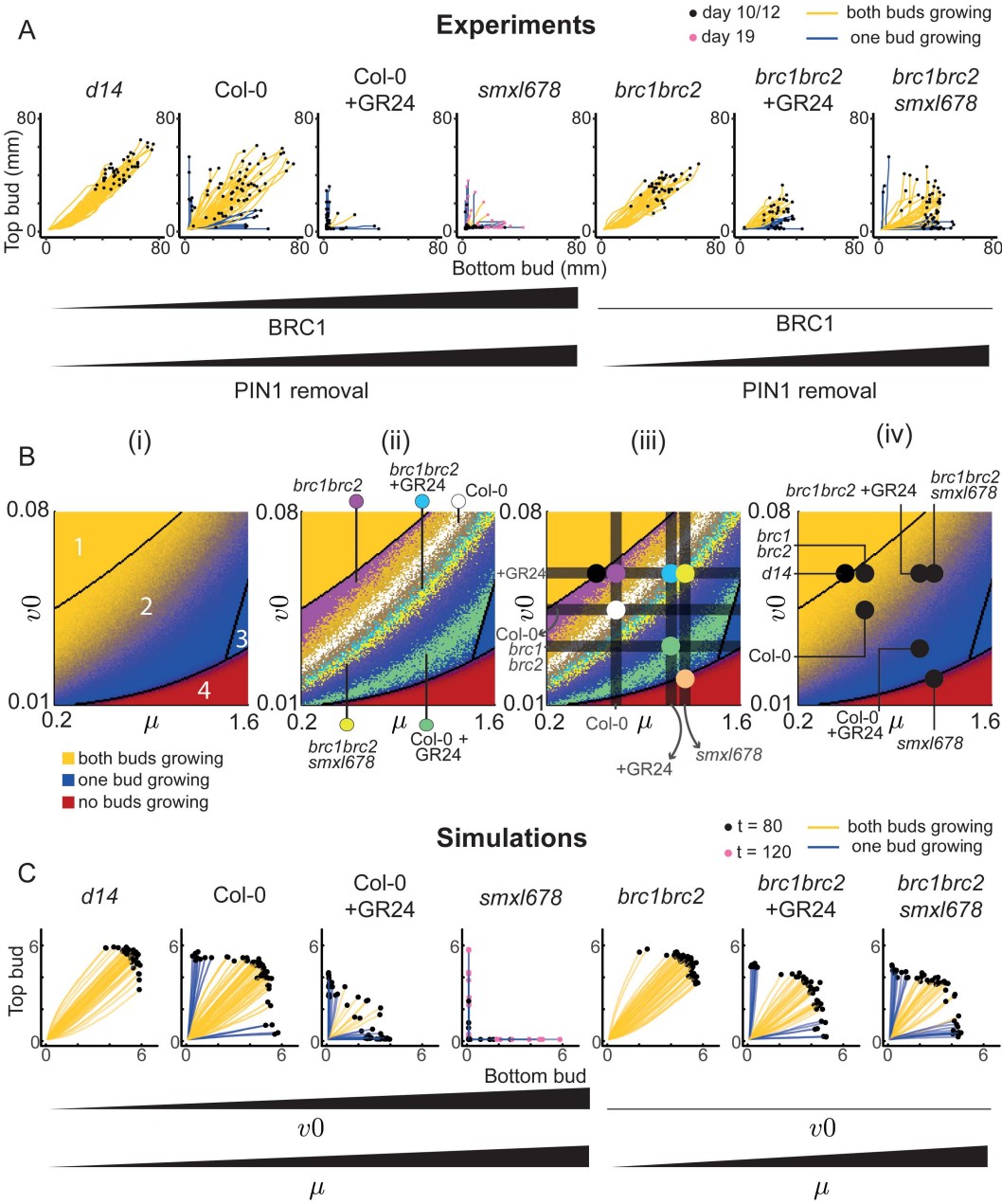

**Fig 4. Variation in two model parameters, representing *BRC1* expression and PIN1 removal, captures variation in bud growth outcomes in 2-node explants for a range of genotypes and treatments. (A)** Mitchison plots with traces colored by growth outcome (yellow-both bud active, blue-one bud active), across genotypes and treatments with known effects on *BRC1* expression and/or PIN1 removal. Buds were measured daily for 10−12 days, except *smxl678* which was measured for 19 days due to its slow bud activation, *n* = 33–94. **(B)** (i) stochastic color map showing the change in growth outcomes across one ($v0$, $\mu$) slice of parameter space, made from a grid of 159 × 159 ($v0$, $\mu$) parameter combinations. The values of the other parameters remain constant. Each pixel represents the bud growth outcome for a combination of ($v0$, $\mu$) values. Black boundaries distinguish regions with a different number and/or value of stable steady states. In regions with two possible behaviors (both buds growing (yellow) or only one bud growing (blue) in region 2; only one bud growing or neither bud growing (red) in region 4), the pixel intensity is calculated based on the relative occurrence of each outcome, obtained from 50 stochastic simulations. **(ii)** First step of the data-fitting process for 2-node data from seven known lines with variations in PIN1 removal and/or *BRC1* expression, mapped as variations in $\mu$ and/or $v0$, respectively. For each genotype/treatment, there is a distribution of points in region 2 (tristable region) where the frequency of bud growth outcomes in the data matches the model. This distribution is superimposed onto the color map from panel **(i)**. The distribution for each genotype/treatment is plotted in a different color, indicated next to the label for each genotype/treatment, and these colors do not relate to the ones used to categorize the bud growth outcome in **(i)**. **(iii)** Superimposed dark grey horizontal and vertical

bars were positioned based on experimental evidence for *BRC1* expression and PIN1 removal in each genetic background/treatment, which constrained the position of each genotype/treatment in parameter space. A colored dot was placed at the intersection of these parameter-constraining bars to assign parameters to each mutant/treatment. **(iv)** Same color map as in panel **(i)**, now with these locations of the genotypes and treatments of interest across the ($v0$, $\mu$) slice. **(C)** Simulated Mitchison plots of genotypes and treatments with variation in $v0$ and $\mu$, the values of which were assigned in panel B, $n$ = 50. Traces are colored according to their growth outcome. Simulations were run for 120 time steps for *smxl678* parameters and 80 for the others. Data and scripts of simulations underlying this figure can be found at https://doi.org/10.17863/CAM.120831.

Overall, the location of each genotype/treatment is on or near the distribution of points where the bud growth outcomes in the experimental data match those of the model. The resulting color map in Fig 4B (iv) shows the narrowed-down regions of the genotypes and treatments in parameter space. The simulations that generate these bud growth outcomes give similar Mitchison plots (Fig 4C) to the experimental data (Fig 4A), and produce associated bud growth dynamics, which we next compared to our experimental data.

## 2.5. The model captures the effects of strigolactone on the duration of the lag phase during bud activation

We previously characterized bud activation as occurring in two phases: a slow-growing lag phase followed by a period of rapid outgrowth [8] (Fig 1C). The parameters chosen to simulate our genotypes and treatments based on bud growth outcomes (Fig 4B) provide predictions about the length of the lag phase and the maximum growth rate of active buds in each genotype/treatment for these parameters.

To test these predictions, we produced a heatmap for the duration of the lag phase in the same ($v0$, $\mu$) slice of parameter space used to map the genotypes/treatments in Fig 4B (Fig 5A). The genotypes and treatments in our experimental dataset were assigned the same $v0$ and $\mu$ values as in Fig 4B. The number of lag days for each set of ($v0$, $\mu$) values are plotted in Fig 5B. In our model, a combined increase in $\mu$ and decrease in $v0$ progressively increase the length of the lag phase across *d14*, Col-0, Col-0 + GR24, and *smxl678*, which spans increasing strength in strigolactone signaling (Fig 5A and 5B, blue data points). This successfully captures the changes in our experimental dataset, where increasing *BRC1* expression and PIN1 removal together progressively delay bud activation (Fig 5C, blue data points).

In the model, an increase in $\mu$ alone causes only a small delay in bud activation (Fig 5B, black data points). This matches our experimental observation that increasing PIN1 removal in a *brc1brc2* mutant background, either through application of GR24, or in a *smxl678* background, has respectively no or a small significant effect on the number of lag days (Fig 5C, black data points). Overall, the model captures qualitatively the observed trends in number of lag days between genotypes and treatments. It is notable that we observe significant within and between-experiment variation in the timing of bud activation in *smxl678* lines (if bud activation is even observed within the timeframe of our experiment) (Figs 2E and 5C). Interestingly, simulated lag days at very low $v0$, representing *smxl678*, are sensitive to small changes in parameter values (S4A and S4B Fig). Trends across our other genotypes/treatments are robust to such changes (S4A and S4B Fig).

## 2.6. The model captures the effects of strigolactone on bud maximum growth rate

After the lag phase, buds switch into a period of rapid outgrowth [8] (Fig 1C), which can be quantified by measuring the maximum growth rate. We generated a heatmap of the maximum growth rate across the previously used ($v0$, $\mu$) slice (Fig 6A). The parameter $\mu$ has a stronger influence on the maximum growth rate than $v0$, because it has a stronger influence on the steady state levels of auxin efflux (S1 Fig), which in the model, determines the growth rate of active buds (Fig 3C). The plots of maximum growth rate for each simulated genotype/treatment show the expected decrease in maximum growth rate with increasing $\mu$ with or without simultaneous changes in $v0$ (Fig 6B). These trends are robust to small changes in parameter values (S4B and S4C Fig).

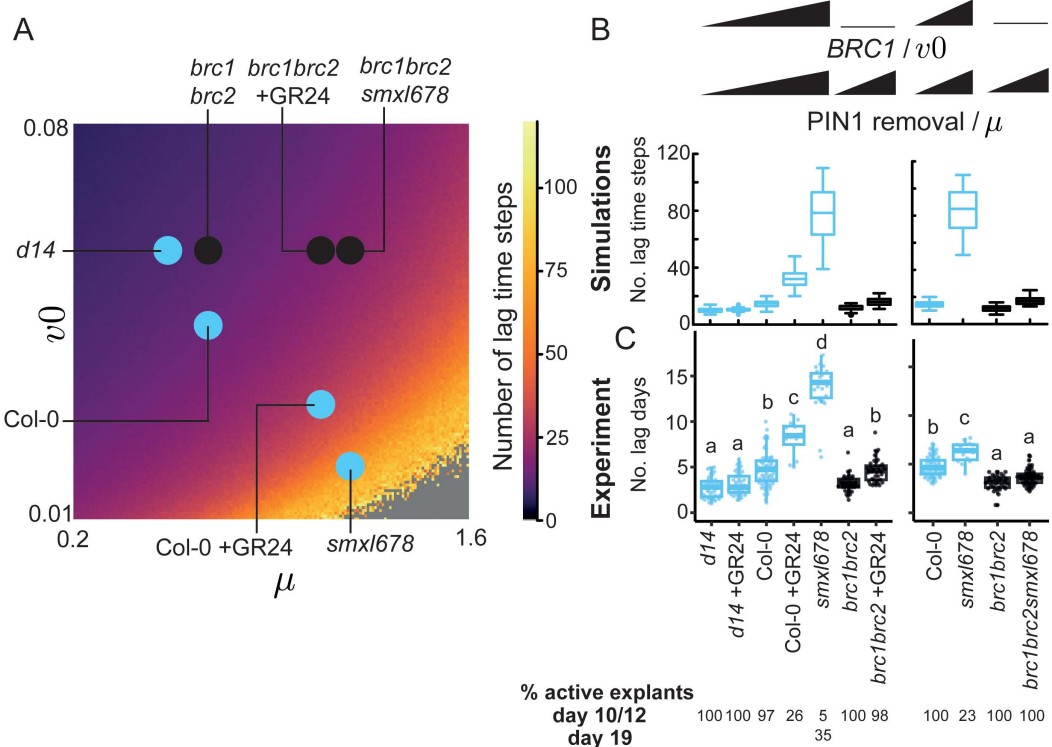

**Fig 5. Variation in two model parameters, representing *BRC1* expression and PIN1 removal, captures variation in the duration of the bud activation lag phase on 2-node explants for a range of genotypes and treatments. (A)** Heatmap of the number of lag time steps for 25,281 combinations of parameters *v0* and *μ* (159 × 159 grid). The pixel color intensity reflects the mean number of lag days from 50 simulations run for 120 time steps. The genotypes/treatments are assigned the same parameter values as in Fig 4B (iii and iv). **(B)** Boxplot of the simulated number of lag time steps calculated for each of the genotypes indicated in panel **A**. The number of lag times steps was calculated from 100 simulations run for 120 time steps. **(C)** Number of lag days calculated as the number of days before growth rate > 2.5 mm/day for the same dataset as Fig 4A. In the left hand plot, buds were measured for 12 days, except *smxl678* which was measured for 19 days. In the right hand plot, all buds were measured for 10 days, *n* = 12–72. Below, the percentage of active explants, defined as explants with at least one active bud, *n* = 43–74. The series of genotypes/treatment with a gradual increase in *BRC1/v0* and PIN1 removal/*μ* are shown in light blue. Data and simulations are shown for the longest bud of active explants. Letters represent statistically significant differences for *p* < 0.05 from a multi-level model with pairwise comparisons and Bonferroni correction. Data and scripts of simulations underlying this figure can be found at https://doi.org/10.17863/CAM.120831.

To test whether this reflects the experimentally observed changes in maximum growth rate, we calculated the maximum growth rate across the same group of genotypes and treatments used previously (Fig 6C). Simultaneously increasing *BRC1* expression and PIN1 removal lowers the maximum growth rate, as can be seen across the *d14*, Col-0, Col-0 + GR24 strigolactone signaling gradient (Fig 6C, blue data points). One discrepancy between the model and our experiments is that *brc1brc2smxl678* had the same maximum growth rate as Col-0 in the experimental data, while the model predicts a lower maximum growth rate (Fig 6B and 6C). The lower maximum growth rate was observed when *brc1brc2* 2-node explants are treated with GR24 (Fig 6B and 6C), indicating the model better captures the maximum growth rate of GR24 treatment than that of a *smxl678* background.

## 2.7. The middle region of the hydrophilic loop is key for strigolactone-mediated effects on PIN1 but not on *BRC1*

We have shown that changes in *v0* and *μ* can capture a wide range of experimentally observed bud growth characteristics across different genotypes and treatments with altered levels of *BRC1* expression and/or PIN1 accumulation. While we can use the *brc1brc2* mutant to test the *BRC1*-independent effects of strigolactone on shoot branching, testing the

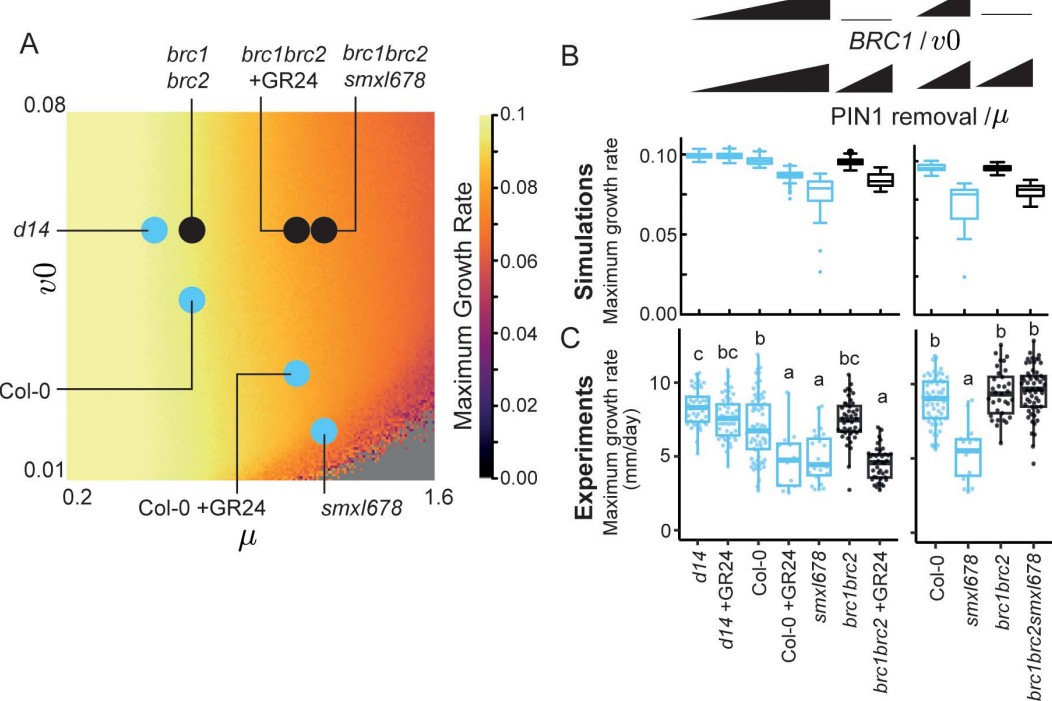

**Fig 6. Variation in two model parameters, representing *BRC1* expression and PIN1 removal, captures variation in the maximum growth rate of buds on 2-node explants for a range of genotypes and treatments. (A)** Heatmap of the maximum growth rate for 25,281 combinations of parameters $v0$ and $\mu$ (159 × 159 grid). The pixel color intensity reflects the mean maximum growth rate from 50 simulations run over 120 time steps. The genotypes/treatments are assigned the same parameter values as in Fig 4B (iii and iv). **(B)** Simulated maximum growth rate for each of the genotypes indicated in panel **A**. The maximum growth rate was calculated from 100 simulations run for 120 time steps. **(C)** maximum growth rate calculated from the same dataset as Fig 4A. In the left-hand plot, buds were measured for 12 days, except *smxl678* which was measured for 19 days. In the right-hand plot, all buds were measured for 10 days, $n$ = 12–72. The series of genotypes/treatment with a gradual increase in *BRC1/v0* and PIN1 removal/$\mu$ are shown in light blue. Data and simulations are shown for the longest bud of active explants. Letters represent statistically significant differences for $p < 0.05$ from a multi-level model with pairwise comparisons and Bonferroni correction. Data and scripts of simulations underlying this figure can be found at https://doi.org/10.17863/CAM.120831.

PIN1-independent effects has so far not been possible, due to the pleiotropy of the *pin1* mutant phenotype, which includes failure to form lateral organs and axillary meristems. To overcome this limitation, we sought to generate a strigolactone-insensitive PIN1 by identifying the region of PIN1 responsible for its strigolactone sensitivity and replacing it with the cognate region from a strigolactone-insensitive PIN family member. Canonical PIN proteins consist of 10 helices functioning as 2 groups of 5 transmembrane domains, separated by a hydrophilic loop exposed to the cytosolic environment. The hydrophilic loop contains several phosphosites which are targeted by kinases to regulate PIN recycling, transport activity, and polarization [62,63], thereby influencing the properties of the auxin transport network. We hypothesized that the hydrophilic loop was a likely target for strigolactone-mediated PIN1 endocytosis.

To test this hypothesis, we performed domain swaps between sections of the PIN1 hydrophilic loop and equivalent regions of PIN3, which is insensitive to strigolactone (Fig 7A, [52]). In particular, we swapped the middle region of the loop, which we termed L2, spanning residues 253–364 of PIN1. To assess the strigolactone sensitivity of the domain-swap constructs, we quantified the effect of GR24 on the accumulation of the chimeric PINs at the plasma membrane of xylem parenchyma cells. Transverse stem sections were incubated in 5 µM GR24 or mock treatment for 6 h before imaging. As expected, GR24 treatment reduced the accumulation of PIN1:GFP at the plasma membrane but did not affect that of

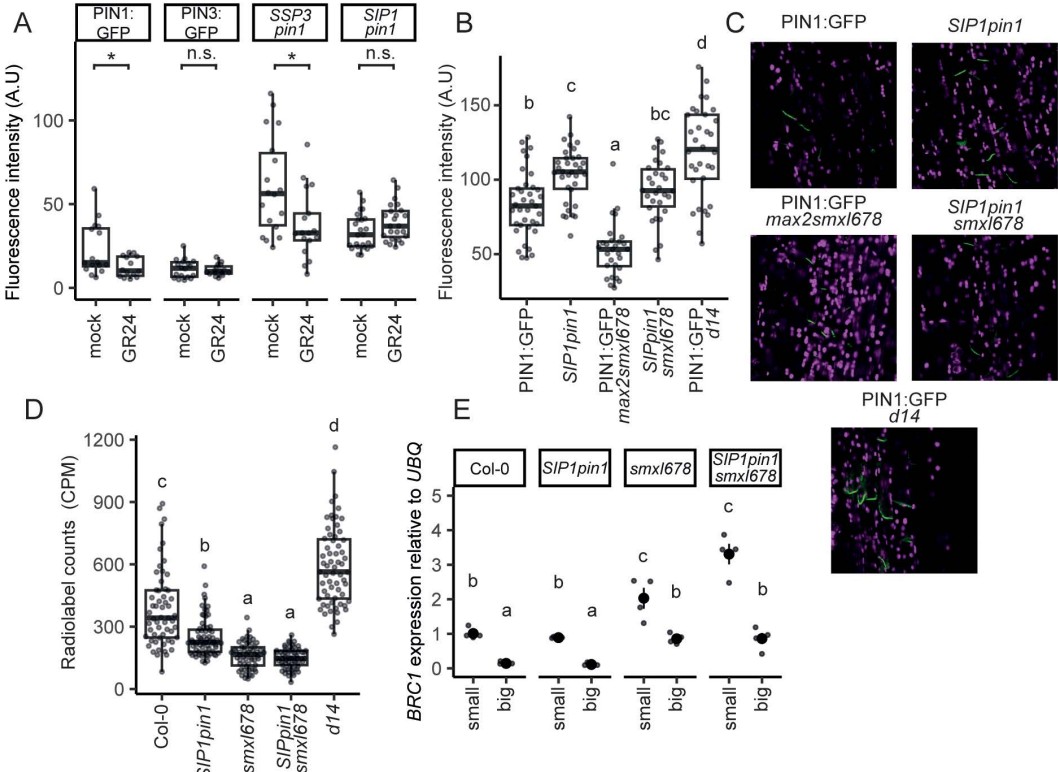

**Fig 7. The middle region of the PIN1 hydrophilic loop confers strigolactone sensitivity. (A)** Fluorescence intensity of basal plasma membranes of xylem parenchyma cells of stems expressing GFP-tagged PIN variants after 6 h mock treatment or treatment with 5 µM GR24. For each stem, the mean fluorescence intensity represents the mean of the 5 brightest membranes for that stem, *n* = 14−24. **(B)** Fluorescence intensity of basal plasma membranes of xylem parenchyma cells for stems expressing GFP-tagged PIN variants of each genotype. For each stem, the mean fluorescence intensity represents the mean of the 5 brightest membranes for that stem, *n* = 28−37. **(C)** Representative images for dataset presented in panel **B**. PIN1:GFP visible in green and chlorophyll autofluorescence in magenta. **(D)** Bulk stem auxin transport in basal inflorescence stem internodes of plants at terminal flowering for the genotypes indicated. Transport was determined as accumulation of radiolabeled auxin in the basal 5 mm of 15 mm stem segments after 16 h incubation with apical 22 nM 3H-IAA, *n* = 56−68. **(E)** Relative expression of *BRC1* as measured by qRT-PCR in Col-0, *SIP1pin1*, *smxl678*, *and SIP1pin1smxl678* in small buds (>2.5 mm, presumed inactive) and big buds (<5 mm, presumed active). Buds were harvested from cauline nodes of whole Arabidopsis plants. Each point represents a biological replicate, with the larger point and bar, representing the mean and associated standard error. Each biological replicate corresponds to the expression level measured in at least 10 pooled buds. Statistical comparisons were made on log transformed data, using two-way ANOVA with Bonferroni correction. In all panels, stars or different letters indicate statistically significant differences at *p* < 0.05 from custom contrasts or pairwise comparison, respectively. Unless otherwise stated, statistical analysis involved a multi-level model with Bonferroni corrections. Data and scripts of simulations underlying this figure can be found at https://doi.org/10.17863/CAM.120831.

PIN3. The fluorescence intensity of the PIN1::PIN3-PIN1$_{L2}$:GFP construct, i.e., GFP-tagged PIN3 with the middle region of PIN1 driven from the *PIN1* promoter, was lower in the GR24 treatment than in mock treated stems, indicating that it is strigolactone sensitive. We named this construct *STRIGOLACTONE SENSITIVE PIN3* (*SSP3*). Conversely, the fluorescence intensity of PIN1::PIN1-PIN3$_{L2}$:GFP, i.e. GFP-tagged PIN1 with the middle region of the PIN3 loop, was the same in mock and GR24 treatments, indicating that this construct has reduced strigolactone sensitivity compared to native PIN1 (Fig 7A and 7B). We named it *STRIGOLACTONE INSENSITIVE PIN1* (*SIP1*). Overall, these results suggest that the L2 region of the PIN1 loop is necessary and sufficient for sensitivity to strigolactone-mediated removal from the plasma membrane.

Crossing SIP1 into the *pin1* mutant background rescued the formation of leaves, axillary buds and flowers (S5A Fig). To assess the shoot branching phenotype conferred by *SIP1pin1*, we quantified branching in plants grown to terminal

flowering in long day conditions (S5B and S5C Fig). *SIP1* confers a modest increase in branching in both the *pin1* and *smxl678pin1* backgrounds, consistent with its strigolactone resistance. However, branching was not significantly increased in the *brc1brc2* background, such that the combination of *brc1brc2* with *SIP1* was unable to phenocopy the full branching phenotype of the *d14* strigolactone receptor mutant.

To assess further the effect of substituting PIN1 L2 for PIN3 L2, we quantified its accumulation on the plasma membrane of xylem parenchyma cells in transverse stem sections in several genetic backgrounds. The fluorescence intensity of SIP1 on *SIP1pin1* membranes was higher than that of PIN1:GFP in a *pin1* background, and its accumulation was not significantly reduced in the *pin1smxl678* background, which has very low accumulation of PIN1:GFP, even with the inclusion of *max2,* to which *smxl678* fully epistatic (Fig 7B and 7C) [14]. These results are consistent with strigolactone insensitivity of SIP1. However, accumulation of SIP1 in a wild-type background was not as high as that of PIN1:GFP in the *d14* background. It is therefore possible that SIP1 retains some strigolactone response.

To test the influence of SIP1 on auxin transport, we performed bulk auxin transport assays through *SIP1pin1* and *SIP-1pin1smxl678* stem segments (Fig 7D). *SIP1pin1* stems have significantly lower levels of bulk auxin transport than Col-0, despite the higher fluorescence intensity on the plasma membrane, and therefore presumably higher membrane accumulation of the protein (Fig 7B). Similarly, high SIP1 accumulation on the plasma membrane did not correlate with high bulk auxin transport activity in the *smxl678* background, with auxin transport levels as low in *SIP1pin1smxl678* as in *smxl678*.

Overall, these results confirm that strigolactone sensitivity of SIP1 is severely compromised, but also suggest that swapping the middle region of the PIN1 loop with that of PIN3 has other impacts beyond changes in strigolactone sensitivity, including impacts on the efficiency of auxin transport. This means that any effects of SIP1 could be due to its strigolactone resistant accumulation, or to these other changes. Furthermore, it is possible that some sensitivity to strigolactone is retained in SIP1.

To test whether SIP1 has any effect on *BRC1* expression, we performed qRT-PCR on RNA extracted from pools of small (presumed dormant) and big (presumed active) buds of Col-0, *SIP1pin1*, *smxl678*, and *SIP1pin1smxl678*. Consistent with previous results and its role suppressing bud growth, *BRC1* expression in Col-0 was higher in small than large buds (Fig 7E). This trend was observed in all lines tested. In addition, as previously reported [11], *smxl678* mutant buds had higher *BRC1* expression than Col-0. The expression level of *BRC1* in *SIP1pin1* and *SIP1pin1smxl678* was not consistently different to that in the cognate native PIN1 backgrounds (Figs 7E and S6). This suggests that SIP1 does not affect *BRC1* expression, consistent with a model where strigolactone-mediated effects on *BRC1* expression and PIN1 removal are independent.

## 2.8. The effect of strigolactone in *SIP1pin1* backgrounds can be predicted by changes in the parameter *v0*

We sought to predict the 2-node bud growth phenotypes conferred by *SIP1pin1.* We account for the lower bulk auxin transport in *SIP1pin1* lines (Fig 7D), by assigning a lower value of S, a parameter corresponding to the efficiency of feedback-driven auxin efflux. This places the *SIP1pin1* mutants on a different (*v0*, *μ*) slice than that used in the previously described simulations (Fig 8A). Predicting the location of *SIP1pin1* lines on this slice (and hence their phenotype) required estimating their values of *v0* and *μ*. Given no evidence of an effect of *SIP1* on *BRC1* expression (Figs 7E and S6), we assigned the *v0* value previously used for each genetic background and treatment. The second parameter *μ*, which corresponds to PIN1 removal from the plasma membrane, is not measured directly. Given that the strigolactone response of SIP1 is reduced compared to PIN1, and the steady state level of GFP accumulation on the plasma membrane is high, we assigned to the *SIP1pin1* lines the same value of *μ* as *d14*, which has no strigolactone-mediated removal of PIN1 (Fig 8A). If in *SIP1* strigolactone acts only via *BRC1*, and not via PIN1 removal, we predicted that *SIP1pin1*, *SIP1pin1* + GR24, and *SIP1pin1smxl678* should have the same low value of *μ*, but progressively lower values of *v0*.

Looking at bud growth outcomes, the model predicts a gradual increase in the proportion of explants with only one active bud across the *SIP1pin1, SIP1pin1* + GR24, *SIP1pin1smxl678* strigolactone signaling gradient (Fig 8A and 8B),

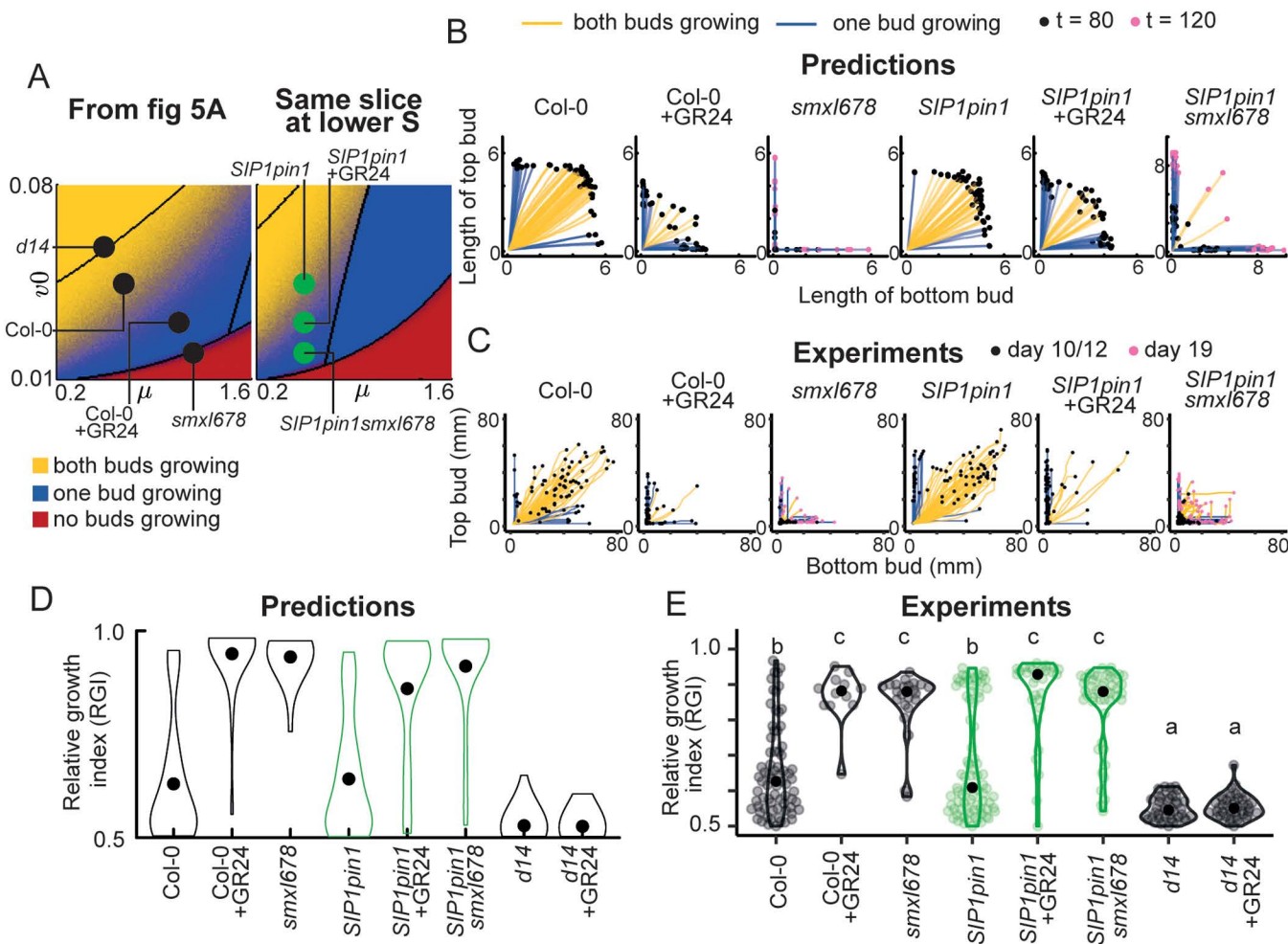

**Fig 8. Model predictions for *SlPpin1* bud growth outcomes largely match experimental data. (A)** Stochastic color map of the bud growth outcomes for the ($v0$, $\mu$) slice used previously (Fig 4), with parameter values for genotypes and treatments in black (as in Fig 5A); and ($v0$, $\mu$) slice at a 20% lower value of the parameter $S$, to account for the lower bulk auxin transport in *SlPpin1* lines, with the predicted location in parameter space of each *SlP1* line shown with a green dot. **(B)** Mitchison plots of predicted (top) and **(C)** empirical (bottom) bud growth phenotypes. Predictions show 50 stochastic simulations per genotype/treatment run for 80 time steps, and those of *smxl678* run for 120 time steps. For the experimental data, *SlPpin1smxl678* buds were measured for 19 days and the other genotypes were measured for 12 days. **(D)** Predicted and **(E)** empirical relative growth index (RGI) for the genotypes/treatments indicated. Data and simulations of Col-0, Col-0 + GR24, and *smxl678* are the same as those from Fig 4A and 4C. The RGI is obtained by calculating the length of the longest bud divided by the combined length of both buds, and is shown only for active explants (those with at least one active bud), $n = 50$ for predictions and 12–80 for experimental data. Letters indicate statistically significant differences at $p < 0.05$ obtained from multi-level model with Bonferroni corrections. Data and scripts of simulations underlying this figure can be found at https://doi.org/10.17863/CAM.120831.

and thus an increasing RGI (Fig 8D). To test this prediction, we performed 2-node assays on these genotypes, including Col-0, Col-0 + GR24 and *d14* as controls. From the Mitchison plots, we indeed found a gradual increase in explants with only one active bud across *SlPpin1*, *SlPpin1* + GR24, and *SlPpin1smxl678* (Fig 8C), which was reflected in the increased RGI (Fig 8E). In *SlPpin1smxl678*, one bud often dominated then stopped growing, followed by the activation of the second bud (Fig 8C), leading to a low RGI despite strong mutual inhibition in this genotype (Fig 8E). Like *smxl678*, *SlPpin1smxl678* buds activated very slowly, so to track their growth we measured bud length for 19 days, while the other genotypes were measured for 12 days.

To assess bud activation dynamics, we produced a heatmap (Fig 9A) and boxplot (Fig 9B) of the number of lag days from model simulations using the parameters assigned for each genotype, with associated sensitivity analysis (Fig 5E and 5F). The simulations show that modelling SIP1 with low $\mu$, wild-type lag days are expected. Gradually increasing $v0$ across the *SIP1pin1*, *SIP1pin1* + GR24, *SIP1pin1smxl678* series, predicts that bud activation is increasingly delayed. The experimentally observed number of lag days calculated from the data validates these predictions (Fig 9C), with a longer delay in bud activation of *SIP1pinsmxl678* than predicted. For the maximum growth rate, the simulations with low $\mu$, produce a wild-type maximum growth rate, and gradually increasing $v0$ across the *SIP1pin1*, *SIP1pin1* + GR24, and *SIP1pin1smxl678* series predicts no effect on maximum growth rate (Fig 9D and 9E). While the comparisons of *SIP1pin1* with *SIP1pin1* + GR24, and *SIP1pin1* + GR24 with *SIP1pin1smxl678*, are consistent with these predictions, we observe a statistically significant decrease in maximum growth rate between Col-0 and *SIP1pin1smxl678* (Fig 9F).

Overall, we find that changes in $v0$ only were largely sufficient to capture the phenotypes we observed across three *SIP1pin1* backgrounds with increasingly strong strigolactone signaling. This is consistent with our hypothesis that PIN1-independent strigolactone action occurs via $v0$, with variation in $v0$ corresponding to BRC1-mediated strigolactone signaling.

## 2.9. Strigolactone acting independently of PIN1 L2 and BRC1 can affect bud growth outcomes and maximum growth rate but cannot delay bud activation

If *SIP1pin1* is fully strigolactone resistant and strigolactone acts only via PIN1 L2 and *BRC1*, we would predict that *SIP1pin1brc1brc2* should have no strigolactone response. To test this prediction, we generated the *SIP1pin1brc1brc2* mutant. The *brc1brc2* mutant background did not affect *SIP1* membrane accumulation or bulk auxin transport as compared to *SIP1pin1* (S7 Fig).

We performed a 2-node assay to compare the effect of GR24 on *SIP1pin1brc1brc2* to that on *SIP1pin1* and *brc1brc2* (S8 Fig). With respect to the lag phase, GR24 extends the number of lag days in both the *brc1brc2* and *SIP1pin1* backgrounds. However, there was no significant effect of GR24 on the lag phase in the *SIP1pin1brc1brc2* background.

In contrast, we found that *SIP1pin1brc1brc2* buds retain a GR24 response with respect to RGI and maximum growth rate, with a small but significant increase observed in both measures (S8A, S8B, and S8D Fig). These results are consistent with SIP1 retaining some strigolactone sensitivity, or with a PIN1-, BRC1-independent mode of action for strigolactone. To assess whether this could be due to the presence of functional PIN7, which is also likely to be strigolactone responsive [52], we made *SIP1pin1pin7*. We observed the same branching phenotype in *SIP1pin1pin7* as for *SIP1pin1* (S5B and S5C Fig). Buds in 2-node explants showed no difference in the responses to strigolactone in the *SIP1pin1pin7* background (S8 Fig). Therefore, the remaining strigolactone responsiveness is unlikely to be mediated by PIN7, but analysis of the *SIP1pin1pin7brc1brc2* would be required to rule this out definitively.

## 3. Discussion

### 3.1. Multi-scale signal integration in the regulation of bud activity

Plants continuously integrate local and systemic inputs to tune shoot branching. Two regulatory hubs have been identified as central to this process, with the BRC1 transcription factor acting locally in the bud to repress outgrowth, and the auxin transport network systemically mediating competition between apices.

The competition between apices is hypothesized to occur via an autocatalytic feedback between auxin transport and its upregulation and polarization in the prevailing transport direction. This creates competition for access to auxin transport in the main stem, with reinforcement through the autocatalytic feedback. Competition with reinforcement is a classical regulatory architecture for a self-organizing system [64,65]. Systemic modulation of the feedbacks in the auxin transport

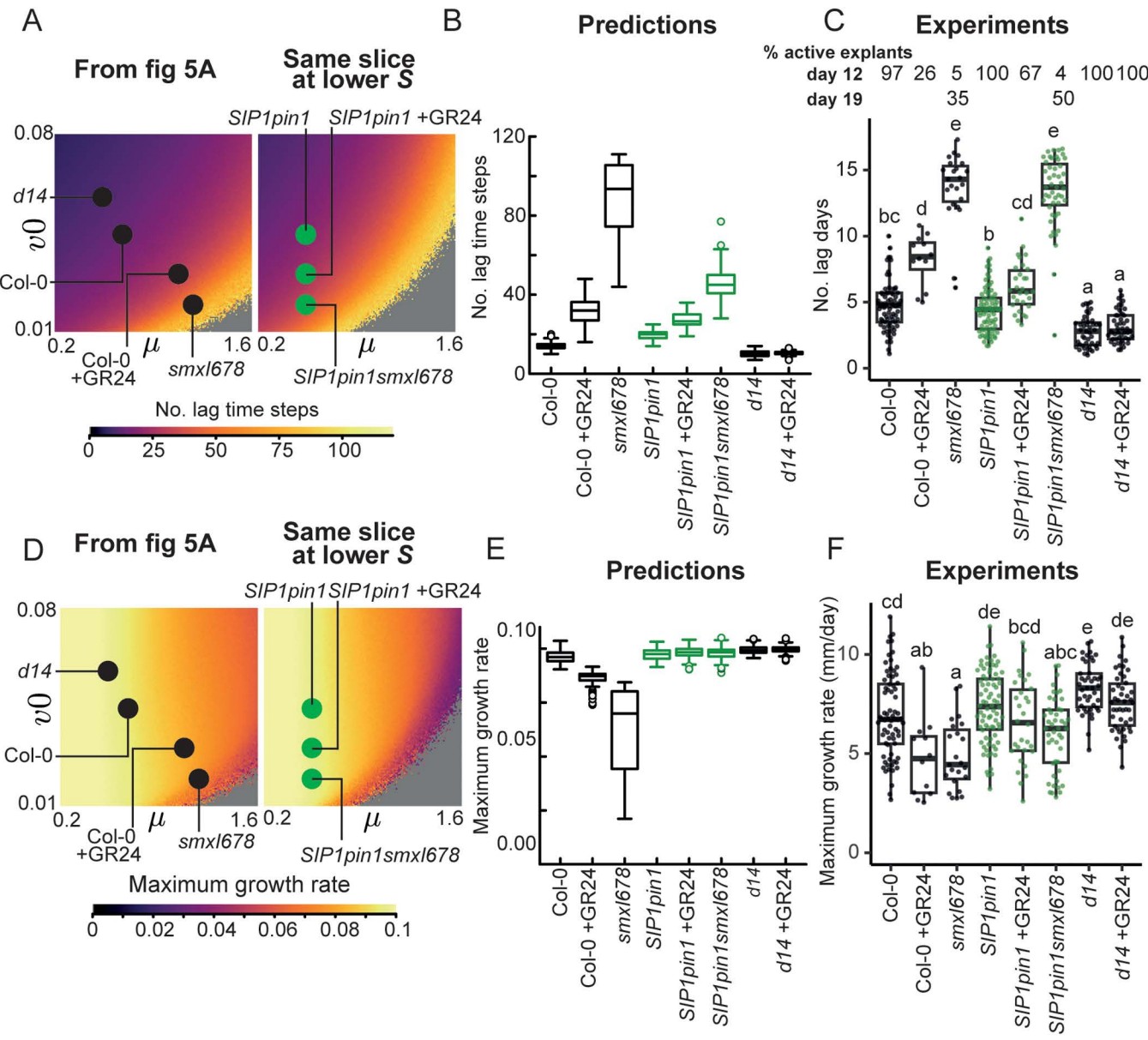

**Fig 9. Model predictions for *SlPpin1* bud activation dynamics largely match experimental data. (A)** and **(D)** Heatmaps of the number of lag days and maximum growth rate from the (*v0*, *μ*) slice used previously (Fig 4), with parameter values for genotypes and treatments from Fig 5A and 6A in black (left); and (*v0*, *μ*) slice at a 20% lower value of parameter S to account for the lower bulk auxin transport in *SlP1pin1* lines, with the predicted parameter values for *SlP1pin1* lines shown in green. Heatmaps are obtained by calculating the number of lag time steps or maximum growth rate for 25,281 combinations of parameters *v0* and *μ* (159 × 159 grid). **(B)** and **(E)** Predictions for the lag time steps and maximum growth rate for genotypes and treatments using the parameter values in **A and D**, each obtained from 100 simulations run for 120 time steps. **(C)** and **(F)** Experimentally determined number of lag days and maximum growth rate calculated from 2-node explants in Fig 8B, where bud length was measured over 12 days, or 19 days for *smxl678* and *SlP1pin1smxl678*, n = 12–80. The percentage of active explants is calculated as the percentage of explants with at least one active bud, n = 43–96. Letters indicate statistically significant differences at *p* < 0.05 obtained from multi-level models with Bonferroni corrections. Data and scripts of simulations underlying this figure can be found at https://doi.org/10.17863/CAM.120831.

PLOS Biology

network allows the number of active apices across the shoot system to be tuned, for example according to nutrient availability [66–68], while exactly which buds activate can be influenced by local bud-acting factors such as light quality.

Local bud competitiveness is likely mediated in part by the BRC1 regulatory hub [5,18,22]. How these local and systemic factors are integrated is not well understood. The analysis of the impacts of strigolactone signaling on shoot branching has provided important insights. Strigolactone acts on both hubs, through upregulating *BRC1* transcription in buds [4,9,10], and through triggering rapid depletion of PIN1 from the plasma membrane in a transcription-independent manner [51], which makes auxin transport canalization more difficult for buds to achieve. This is consistent with the high branching of strigolactone receptor or biosynthesis mutants [69–71].

Further insights into the regulation of bud activity come from considering the impacts of mutations and treatments on the slow-growing lag and rapid growth phase in isolated 1-node explants [46,60,72]. For example, strigolactone extends the lag phase during bud activation and reduces the maximum growth rate of active branches, and *brc1brc2* mutant buds are relatively strigolactone-resistant to effects on the former but not on the latter [8]. This approach has enabled the dissection of the shoot branching regulatory network. It has provided evidence to support the ideas that (i) the establishment of auxin transport canalization occurs during the slow-growing lag phase of bud activation, with the transition to rapid growth corresponding to fully canalized export, which is hard to reverse, (ii) BRC1 influences the length of the lag phase, and (iii) the rate of rapid growth is influenced by the level of auxin efflux from the bud.

Complementing results from isolated 1-node explants, 2-node explants are a powerful minimal system to study the impact of another bud on growth dynamics, thereby revealing some of the systemic properties of shoot branching regulation [30,32,45,53]. After excision from the plant, typically both buds on 2-node explants enter the slow-growing lag phase. If both buds have strong self-activation and/or low mutual-inhibition, then both activate. Alternatively, if one bud establishes rapid growth quickly and inhibits the growth of the other, then the second bud will stop growing. This phenomenon can be quantified using the relative growth index (RGI). Even when both buds activate, the presence of another bud extends the lag phase of both buds, and reduces their maximum growth rate [8]. Strigolactone has the same effects on the lag and rapid growth phases as in 1-node explants (Figs 5 and 6), and additionally increases the RGI, typically resulting in one bud dominating the other [51]. Hence analyzing the effect of strigolactone at the scale of 2- rather than 1-node explants casts strigolactone as systemically tuning the level of bud-bud competition rather than simply inhibiting bud activity [49,51]. The RGI in *brc1brc2* is also increased by strigolactone, though to a lesser extent than in wild-type [22].

Combining *brc1brc2* with the *smxl678* constitutive strigolactone response mutant further elucidated the interplay between the two hubs in the strigolactone-mediated regulation of bud-bud competition (Fig 2). In 1-node explants, *smxl678* buds are inhibited, while *brc1brc2* and *brc1brc2smxl678* buds have the same rapid bud activation (Fig 2B). However, *brc1brc2* and *brc1brc2smxl678* do not exhibit the same phenotype in 2-node explants, where the level of bud-bud competition is higher in *brc1brc2smxl678* than *brc1brc2* (Fig 2D and 2E). These observations can be explained within a canalization-based model, where high PIN1 removal in *smxl678* and *brc1brc2smxl678* makes canalization harder to establish [14]. Bud activation may be possible despite high PIN1 removal only in more permissive contexts, as seen in 1-node explants of *brc1brc2smxl678* but not *smxl678*. In 2-node explants, high bud-bud competition in *brc1brc2smxl678* may emerge from a combination of low *BRC1* expression, creating a locally permissive environment in each bud, combined with a systemic canalization-inhibitory environment from high PIN1 removal.

Taken together, the suite of assays, metrics, mutants and treatments presented above provide a powerful toolkit to understand the properties of the bud regulatory network, and to explore the relationship between systemic and local regulation, and the two known bud regulatory hubs.

## 3.2. Computational models for understanding bud regulation

In addition to experimental work, computational modelling has been key to providing evidence that the proposed regulatory architecture is capable of delivering the wide range of behaviors and properties we observe. In particular, our previous

model has demonstrated that many non-intuitive bud growth phenomena can be explained under the auxin transport canalization-based model for bud activation, with strigolactone promoting PIN1 removal from the plasma membrane [25,49].

Here we describe an extension of this previous work. We incorporate the key features of the auxin transport canalization-based model for bud activation into a simplified model representing the 2-node branching assay (Fig 3). The model includes two new features based on the advances in our understanding of bud growth dynamics outlined above. First, we explicitly model bud growth rate, expressing it as a function of bud auxin export. Second, we model BRC1 as affecting basal levels of auxin efflux. This latter hypothesis was motivated by the characteristic short lag phase for bud activation observed in *brc1brc2* mutant backgrounds, suggesting rapid auxin transport canalization, along with the ability of the *abcb19* mutant to suppress many of the bud growth phenotypes of *brc1brc2* mutants [8,52]. The ABCB19 auxin efflux carrier is plausibly a major contributor to basal levels of auxin efflux.

Using this simple model, with strigolactone acting through both PIN1 removal and BRC1-mediated reduction in basal levels of auxin efflux, we simulate a wide range of 2-node assay experimental results. We compare these simulations to empirical results, for corresponding mutant backgrounds and treatments. We focus on dissecting the dual activity of strigolactone, using *brc1brc2* mutants and a PIN1 variant that has compromised strigolactone response.

### 3.3. Two biologically relevant model parameters capture multiple properties of bud behavior

Based on the above reasoning, we model *BRC1* as influencing basal auxin efflux via *v0*, and PIN1 removal as *μ*. Tuning these parameters to match the effects of a range of relevant loss of function mutations and strigolactone treatments successfully captures much of the corresponding variation in bud behavior in 2-node assays, including effects on the number of lag days, maximum growth rate, and growth outcomes.

For example, at high *v0* (low BRC1), changes in *μ* (PIN1 removal) modify 2-node growth outcomes without substantial effects on the number of lag days (Fig 5A). This captures our observation that increasing PIN1 removal through strigolactone treatment or *smxl678* mutation in a *brc1brc2* background leads to one bud more often dominating the other, while maintaining rapid bud activation (Fig 5C). Furthermore, an increase in *μ* creates a much greater delay in bud activation at low compared to high *v0*. The effect of one parameter therefore depends on the value of the other, matching the different impacts on the time of bud activation of *smxl678* mutation in *brc1brc2* and wild-type backgrounds.

The effectiveness of this simple model in capturing so many of the behaviors of buds on 2 node explants provides evidence that the hypotheses underlying the model are correct. In particular, the concordance between modelled and observed behaviors provides further evidence to support the canalization-based model of bud activation, with buds competing to establish canalized auxin transport into the stem, and the lag phase corresponding to the canalization period. Furthermore, the results support the two new hypotheses we propose here, namely that *BRC1* downregulates the basal rate of auxin efflux, and that bud growth rate is influenced by the level of auxin efflux.

The model used here replicates key features from the whole plant model set out in Prusinkiewicz and colleagues [25]. This model is able to reproduce the spatial and temporal patterns of bud activation in multiple plant architectures by simple parameter changes. Combining the novel features of the 2-node model presented here with the whole plant model described previously could provide further insights on how growth is balanced across the shoot system.

### 3.4. Mechanisms of BRC1 action

Basal auxin efflux (i.e., auxin efflux that is independent of canalization), occurs principally through the action of non-polar auxin transporters and diffusion, and depends on auxin levels. Hence for BRC1 to reduce basal auxin efflux, it could act in a number of ways.

First, BRC1 could down-regulate basal auxin efflux by downregulating the activity of non-polar auxin transporters, which are collectively expressed more broadly than PIN1. These transporters, namely PIN3, PIN4, PIN7 (collectively

PIN347), and ABCB19 are thought to be important in the early stages of canalization, promoting auxin loading in the bud and background auxin flux between the bud and the stem. This enables the subsequent polarization of PIN1, which drives the high-capacity, highly polar auxin transport characteristic of canalized tissue [52,61,73]. This hypothesis is consistent with the observation that BRC1 influences the length of the lag phase and not the maximum growth rate, because BRC1-mediated regulation of ABCB19/PIN347 would mainly influence the dynamics of auxin flux before PIN1 polarization, regulating the timing of the transport switch but not the steady state levels of auxin transport, which are principally determined by PIN1.

Previous analyses of transcriptional changes downstream of BRC1 do not identify auxin transporters as BRC1 targets in Arabidopsis, although in cucumber, BRC1 has been shown to regulate PIN3 expression [74]. BRC1-dependent regulation of PIN347/ABCB19 could occur post-transcriptionally for example via phosphoregulation, an important mode of non-transcriptional regulation of PIN1 activity [63]. An effect could also occur via ABA. Three downstream targets of BRC1 are HD-ZIP transcription factors promoting ABA accumulation [18,20], and ABA influences PIN2 accumulation in roots [75,76].

Alternatively, BRC1 may down-regulate the basal rate of auxin efflux by tuning the available auxin concentration. BRC1 could lower auxin availability by promoting amino-acid conjugation of auxin via GH3.5, an IAA-amido synthetase involved in auxin homeostasis identified as a direct BRC1 target [18]. In the model, auxin concentration affects both the basal rate of auxin efflux via $v0$ and the maximum rate of the Hill function via $v$. For the parameters investigated, $v0$ is a bigger determinant of early efflux levels, while $v$ acts in the positive feedback of canalization and so becomes increasingly important as auxin efflux increases. Given BRC1 expression is rapidly downregulated during bud activation [4], BRC1-mediated effects on auxin availability would be most prominent early in bud activation, modifying basal auxin efflux, $v0$, without affecting the positive feedback via $v$.

BRC1-mediated regulation of basal auxin efflux would provide a mechanism through which local environmental conditions in the bud, which influence BRC1 expression [19,20], can be integrated into the systemic canalization-based regulation of bud activation. Buds along a stem would exhibit variation in their basal rate of efflux, enabling those in more favorable conditions to establish canalized auxin transport into the stem, while the feedbacks between auxin sink and source strengths would balance the number of active apices across the whole plant.

### 3.5.  PIN1-mediated action of strigolactone

Further tests of our model would benefit from the ability to prevent strigolactone action specifically via PINs. As a first step toward this, we identified the middle region of the PIN1 cytoplasmic hydrophilic loop as necessary and sufficient for strigolactone-mediated removal of PIN1 from the plasma membrane (Fig 7). This allowed us to create SIP1, a chimeric PIN1 in which the middle L2 region of the PIN1 loop has been swapped for that of PIN3. GFP-tagged SIP1 remains on the membrane in the presence of strigolactone and in a smxl678 background, confirming its strigolactone resistance at least with respect to accumulation at the plasma membrane.

It is important to note that SIP1 confers low bulk auxin transport levels (Figs 7D and S7), consistent with the hydrophilic loop being a dense regulatory environment with multiple phosphorylation sites important for PIN1 activity [63], which may have been affected in SIP1. This complicates the interpretation of SIP1 phenotypes. The low auxin transport efficiency of SIP1 might impact all the phenotypes of interest independently of strigolactone-mediated PIN1 removal. For example, the whole plant phenotypes conferred by SIP1 include a modest increase in branching, which contrasts to the highly branched phenotype of brc1brc2. This could indicate that strigolactone signaling via PIN1 has a relatively minor role in determining branching in whole plants. However, it is possible that in whole plants, the low transport efficiency of SIP1 masks the impact of its strigolactone resistance on shoot branching. Identification of the precise residues required for strigolactone-mediated PIN1 removal would further our understanding of strigolactone signaling and the role of strigolactone-mediated PIN1 removal in shoot branching.

Despite this complication, because the compromised auxin transport efficiency was common across all SIP1 lines, we could still investigate the PIN1 L2-independent effect of strigolactone by comparing phenotypes of *SIP1pin1*, *SIP-1pin1* + GR24, and *SIP1pinsmxl678*, which constitute a gradient of increased strigolactone signaling. After accounting for the lower auxin transport efficiency in our model, changes in $v0$ alone could accurately predict the effect of strigolactone in the SIP1 lines/treatment, further supporting our model. For example, the reduced effect of strigolactone on the maximum growth rate in the *SIP1* background is consistent with our hypothesis that growth rate is influenced by the steady state level of auxin efflux, which in the model is not significantly affected when strigolactone acts via $v0$ (BRC1) and not via $\mu$ (PIN removal) (S1 Fig).

A second complication is that SIP1 does not over-accumulate to the same degree as PIN1 in the strigolactone receptor mutant background *d14*. Together with the residual strigolactone sensitivity of RGI and maximum growth rate in *SIP1pin-1brc1brc2*, this leaves open the possibility that strigolactone acts through additional PIN regions, or through an entirely different mechanism in addition to its PIN1 and BRC1-mediated effects, such as via trehalose-6-phosphate [77]. Nonetheless, representing strigolactone action in our model solely via *BRC1* and PIN1 removal was sufficient to capture multiple aspects of bud behavior observed in the range of mutant and transgenic backgrounds and treatments we tested. Importantly, the observation that *SIP1pin1brc1brc2* was resistant to strigolactone-mediated effects on the number of lag days establishes PIN1 L2 and *BRC1* as the two necessary strigolactone targets for its effects on the length of the lag phase.

### 3.6. Discrepancies between model and data in *smxl678 mutant* backgrounds

While there is generally a close match between the model predictions and empirical results, there are also some discrepancies. These occurred particularly in *smxl678* backgrounds. The maximum growth rate of *brc1brc2smxl678* and *smxl678*, and the number of lag days of *SIP1pin1smxl678*, were higher than predicted by the model; and the maximum growth rate of *SIP1pin1smxl678* was lower than predicted by the model. There are several explanations for this. The *smxl678* background may represent fully constitutive strigolactone signaling, and it is possible that the model simply does not adequately capture this extreme state. Alternatively, there may be some SMXL678-independent action of strigolactone, consistent with recent work showing D14 can be degraded independently of SMXL678 [78]. For example, these effects may occur via trehalose-6-phosphate [77,79].

The role of sugar in bud regulation may also account for the counterintuitive observation that *smxl678* buds grow out less often on 1-node compared to 2-node explants, despite the mutual inhibition between buds in the latter case (Fig 2B and 2D). Sugar availability from leaves influences bud release in pea and rose [80,81], and sustained bud growth in Arabidopsis explants [72]. Sugars up-regulate auxin levels in several species and developmental contexts [82,83], including axillary buds of rose [84]. In our mathematical model, higher global auxin concentration increases the basal rate of auxin efflux $v0$ and the maximum rate of the Hill function $v$ (mathematical appendix), thereby facilitating the activation of both buds (S2 Fig). Hence in *smxl678* 2-node explants, higher sugar from the presence of a second leaf might facilitate the establishment of auxin canalization, enabling some degree of bud outgrowth.

### 3.7. Conclusions

We propose a model that integrates *BRC1* and auxin transport, two previously independent hubs in shoot branching regulation. Our model captures and predicts the phenotype of plants with defects in either one or both hubs, at the level of both bud growth dynamics and bud-bud interactions. This supports the model formulation, namely that buds compete to establish canalized auxin transport into the main stem, with the competitive strength of each bud locally tuned by *BRC1*. This regulatory system enables both the number and location of growing branches to be coordinated across the whole shoot system. Future experimental work should continue to investigate the detailed mechanisms underlying these hypotheses. Our model also guided our inquiry into the strigolactone-mediated regulation of shoot branching, leading us to identify a region of the PIN1 hydrophilic loop key for strigolactone responsiveness, which should be studied in more detail

in the future. Experimental findings obtained from these lines of research could be iteratively incorporated back into our model, to continue exploring the multi-scale regulatory logic through which plants tune their shoot architecture.

## 4. Materials and methods

### 4.1. Experiments

**4.1.1. Plant material and growth conditions.** *Arabidopsis thaliana* (Arabidopsis) (L.) Heynh. Col-0 ecotype was used throughout. The following mutants have been described previously: *brc1-2 brc2-1* [4], *d14-1* [85], *smxl6-4 smxl7-3 smxl8-1*, *smxl6-4 smxl7-3 smxl8-1 max2-1*, and *smxl6-4 smxl7-3 smxl8-1 brc1-2 brc2-1* [14].

To generate the chimeric PIN lines, *SIP1* and *SSP3*, the hydrophilic loop of PIN1 and PIN3 were split into 3 regions: L1 from residue 149–253, L2 from 253 to 364, and L3 from 364 to 468. The *SIP1* line was synthesized by generating a chimeric coding sequence (CDS), consisting of the entire PIN1 CDS with the L2 region of PIN1 substituted with that of PIN3, and GFP located in the L3 region [86]. This chimeric CDS was flanked by the attL1 and attL2 sites at the 5′ and 3′ ends, respectively. For the SSP3 construct, the entire *PIN3* coding region, including the *GFP* insertion, was amplified from a previously published reporter construct [87] and cloned into pDONR221. This vector was used as a template to produce the chimeric construct. Coding sequences for the L2 region of the *PIN1* hydrophilic loop were amplified from a published *PIN1-GFP* line that was shown to be able to complement the *pin1* mutant [86]. The L2 region of the *PIN1* loop, and the *PIN3* backbones missing the homologous L2 region, were amplified and recombined using sequence and ligation-independent cloning (SLIC) to generate the chimeric *PIN3* in the pDONR221 [88]. The expression of both constructs was driven by a region previously reported to complement the *pin1* mutant when driving *PIN1* expression [89]. This 3.5 kb region was amplified from plasmid DNA and recombined into Gateway vector pDONR P4-P1R. To generate expression vectors, LR reactions were performed according to manufacturer's instructions, using Gateway binary vector pH7m24GW as the destination. The resulting vectors were transformed into *Agrobacterium tumefaciens* and used to transform *Arabidopsis thaliana* via floral dip [90], and crossed into other mutant backgrounds to generate the homozygous mutant lines as needed.

Plants were grown on Levington F2 compost. Plants and explants were grown in Conviron growth chambers at 22 °C/18 °C and 16 h/8 h day/night cycles of temperature and light, respectively.

**4.1.2. Bud growth assays.** A 1.5 ml lid-less Eppendorf tubes filled with liquid *A. thaliana* salts (ATS) media containing no sucrose [91] were sealed with parafilm. Young Arabidopsis inflorescences were harvested, inserted into a tube and decapitated at day 0. Explants were decapitated above the lowest node and above the lowest two nodes for one and two-node explants, respectively. Bud length was measured manually. The few samples that dried out and wilted before the end of the experiment were discounted from the analysis. For strigolactone treatments, a stock solution of 10 mM GR24 (LeadGen Labs LLC) in 90% acetone was used at a final concentration of 5 µM. Mock solutions contained the same volume of acetone without GR24.

**4.1.3. qRT-PCR.** Pools of at least 10 buds were harvested and immediately frozen in liquid nitrogen or dry ice. RNA was isolated using the Zymo Quick-RNA Miniprep Kit. RNA was quantified with NanoDrop One and 1 µg RNA was used for cDNA synthesis. All transcript levels were quantified relative to *UBQ10*. Relative transcript levels were quantified using the delta–delta-Ct method [92].

**4.1.4. Microscopy.** Stems from 4–5 week old plants (inflorescence height of 5–10 cm) were harvested for imaging. The most basal 15 mm segment was isolated and hand sectioned longitudinally with a razor blade to expose the xylem parenchyma. Bisected stem segments were immobilized with tape then immersed in water and imaged immediately, or immersed in 5 µM GR24 or mock solution for 6 h before imaging. Confocal images were captured on a Zeiss LSM 700 confocal microscope equipped with a 20× water immersion objective. Sample excitation was performed using solid-state lasers, a 488 nm laser at 6.0% laser intensity, and a 639 nm laser at 2.0% laser intensity. Image size was set to 1024 × 1024 pixels, with 4× line averaging. The pinhole was set to 1 airy unit. The fluorescence intensity at the plasma

membrane was quantified using the Zeiss Zen 2012 Software. Polygons were drawn manually around the plasma membrane before calculating the mean intensity of the selected area. The signal intensity for each stem was acquired by averaging the signal from at least 5 individual cells. The images in Fig 7C were cropped to 383 × 383 pixels to improve clarity.

**4.1.5. Auxin transport assays.** For stem auxin transport assays, the most basal 15 mm stem segment of the primary inflorescence, from plants at terminal flowering, was harvested. The apical end was placed into the well of a PCR plate with 22 nM 3-H IAA and 0.005% Triton X-100. Plates were sealed and incubated in the light for 18 h. The basal section of each stem was cut into two 2.5 mm stem segments, both of which were placed in the well of a 96 well plate with 200 µL MicroScint-20 scintillation liquid (PerkinElmer). Samples were shaken for at least 4 h at 450 RPM before scintillation counting.

**4.1.6. Branch quantification.** Primary branching was quantified for each plant at terminal flowering, counting the number of first-order cauline and rosette branches >5 mm.

## 4.2. Statistical analysis

All statistical analysis was carried out in R v.4.1.31 and RStudio v.2022.02.1 + 461. Bud growth data was analyzed by fitting a growth model [93] to each bud growth trace in R, as specified in [8]. The number of lag days and maximum growth rate is only shown for active buds (maximum growth rate > 2.5 mm).

For statistical testing, multi-level models were fitted to the data to account for random variation between experiments. A linear multi-level model was used with either a normal or gamma distribution depending on the data. The model was fit using the R/lme4 package (v1.1.29; [94]) and contrasts of interest extracted using the R/emmeans package (v 1.8.2; [95]). Estimated marginal means and pairwise comparisons between genotypes were adjusted for multiple testing using Bonferroni correction. The box for all boxplots spans the first to the third quartile range with the middle line showing the median, and the superimposed black dot showing the mean. The minimum and maximum whisker values are calculated as Q1/Q3 (first or third quartile) ± 1.5*interquartile range. Adobe illustrator v29.5.1 was used to add the results of statistical tests to the plots and generate multi-panel figures.

## 4.3. Computational modelling

**4.3.1. Model implementation.** The formulation and derivation of the model is detailed in the model appendix. The model was implemented in Julia version 1.7.2. The following packages were used: Catalyst.jl for model generation [96,97], DifferentialEquations.jl for numeric simulations [98], Plots.jl for figures, ImplicitEquations.jl for nullcline calculation, HomotopyContinuation.jl [99] and LinearAlgrebra.jl for steady state analysis. To implement stochasticity, the Catalyst modelling software was used, which automatically generates stochastic differential equations from chemical reaction network models. A noise parameter was added to scale linearly the magnitude of the noise introduced to the Chemical Langevin Equation [97].

**4.3.2. Simulations.** Stochastic differential equations were used to simulate the auxin efflux and growth of 2-node explants. The implicit Euler-mauruana numerical method was used, as previously [97]. We used the implementation from the StochasticDiffEq.jl package [98].

All simulations were performed with identical starting conditions for both buds: 0.01 for $E$ and $F$, and 0.1 for $N$ and $M$. Simulations were run for the number of time steps indicated. Parameters remained constant during the simulations.

**4.3.3. Steady state analysis.** The steady states of auxin efflux were obtained numerically using the HomotopyContinuation.jl package. The solutions were categorized into stable and unstable steady states using a standard method [100]. The steady state analysis was performed on the equations for auxin efflux, but they are also interpreted at the level of bud growth, because, for the parameters of interest, switching from low to high efflux is necessary for bud growth. Model behaviors were categorized based on the number and value of the stable steady states (S2A Fig). A steady

state value of 0.4 was used to threshold between a steady state that corresponds to both buds activating compared to neither buds activating (S2A Fig, behaviors 1 versus 6). In behaviors 2 and 4 (S2A Fig), both buds can reach the same steady state value, but this corresponds to an active and an inactive state, respectively. A threshold value of 0.24 was used to distinguish between these two cases.

**4.3.4. Color maps.** Color maps were obtained by classifying model behavior based on steady state analysis [101]. To generate the non-stochastic color maps (S2B Fig), a pixel value was assigned based on the model behavior for a combination of ($v0$, $\mu$) values. The color map was obtained by plotting the pixel value obtained for each parameter combination.

To generate the stochastic color maps (Figs 4B, 8A, and S3), steady state analysis was performed for a combination of 25,281 pairs of ($v0$, $\mu$) values. In the mono and bistable regions (when both buds grow, only one bud grows, or neither grow), stochasticity does not influence the number of stable steady states reached in a simulation. Parameter combinations leading to only one of these three outcomes were assigned a unique color. In the tristable and quadrastable regions, stochasticity does influence the value of the steady state reached. To represent this effect, in these regions, 100 stochastic simulations were performed for each set of parameters for 120 time steps. The pixel value was calculated based on the growth outcomes in these simulations. For example, if certain parameters lead to 70% of simulations with both buds growing and 30% with only either the top or bottom bud growing, the pixel intensity was calculated as follows: RGB(point) = 0.7 × RGB(yellow) + 0.3 × RGB(blue), where RGB(point) refers to the pixel value (i.e., color) on the color map representing the outcome of competition for specific parameter values. The outcome "one bud growing" was defined as occurring if the length of one bud was more than three times the length of the other. The color map was obtained by plotting the pixel value obtained for each parameter combination. In some rare occurrences, the function that calculates the stable steady state fails. In this case, the pixel was assigned the same value as that of the neighboring pixels if these are all identical, otherwise the pixel was assigned a value equal to the mean of all neighboring values. To visualize the boundaries between the region of parameter space with a different number of stable states, a black color was assigned to all pixels which had more than one neighboring pixel with a different number of stable steady states.

**4.3.5. Heatmaps of number of lag days and maximum growth rate.** A total of 100 simulations were run for 12 time steps for 25,281 combination of parameter values. We then calculate the number of lag days and maximum growth rate for each simulation where buds activated. Buds were considered active if growth rate > 0.02 during the simulation. The number of lag days was calculated as the number of days before growth reaches 0.02. To categorize bud activity and obtain the maximum growth rate, we first smooth the noise from the stochastic simulation then extract the maximum value of the first derivative.

**4.3.6. Sensitivity analysis.** For sensitivity analysis, the steady states and growth outcomes (S3 Fig) or maximum growth rate, number of lag days, and relative growth index (RGI) (S4 Fig) were calculated from simulations with a systematic decrease or increase of 5% in all parameter values [96].

## Supporting information

**S1 Fig. Deterministic simulations showing the influence of each auxin efflux parameter on the timing of the switch and the steady state level of auxin efflux.** Each panel shows the auxin efflux $E$ over time obtained from 5 deterministic simulations performed until time step $t = 100$, changing only one parameter value. Because these are deterministic simulations, the auxin efflux $F$ is identical to that of E. Scripts of simulations underlying this figure can be found at doi 10.17863/CAM.120831.
(TIF)

**S2 Fig. Behavioral classification and scan of parameter space. (A)** The six possible behaviors found across parameter space, illustrated with a Mitchison plot of five simulations in the corresponding area of parameter space. The behaviors

are categorized based on the number of stable steady states and their values. Corresponding nullclines in black and grey show how the shape of the nullclines influences the number of stable steady states, with steady states indicated by a dot at the intersection of the nullclines. Each behavior is represented in a different color: yellow represents cases where both buds activate, green when either both buds activate or only one bud activates, navy when only one bud activates, light blue when either neither bud activates or only one bud, white when all four outcomes are possible (neither bud, both buds, or only one bud), and red when neither bud activates. **(B)** Color maps showing how the ($v0$, $\mu$) slice chosen for further analysis (middle row) is affected by changes in $S$, $D$, $v$, $K$ and $n$. The behavior at each point in parameter space on the color map is indicated by the color of the pixel at that point, as specified in panel **A**. The middle slice was chosen based on its ability to deliver core features of bud activation. The slices above and below show how a respective 50% decrease or increase in each parameter value changes the growth outcomes. For example, at lower D there is a larger area of parameter space where both buds grow (behavior 1 – yellow). Here, colors are assigned based on the theoretically possible behaviors, as determined by the number and values of stable steady states. In Figs 4 and S3, colors are assigned based on the bud growth outcomes that were obtained after running 100 stochastic simulations for 120 time steps, for each combination of ($v0$, $\mu$) values. Scripts of simulations underlying this figure can be found at doi 10.17863/CAM.120831.
(TIF)

**S3 Fig. Sensitivity analysis of chosen slice introduced in** Fig 4B**.** The stochastic color map from Fig 4B (center) is shown with a 5% decrease (left) and a 5% increase (right) applied to parameters $S$, $D$, $v$, $K$, $Q$, and $\eta$. Scripts of simulations underlying this figure can be found at doi 10.17863/CAM.120831.
(TIF)

**S4 Fig. Sensitivity analysis of the lag time steps and maximum growth rate simulations from (A) and (B):** Fig 5C**, (C) and (D):** Fig 6C**, (E) and (F):** Fig 9C **and** 9F**.** In each case, the number of lag days and maximum growth rate was calculated from 100 simulations performed for 120 time steps, with all parameter values increased or decreased by 5%, as compared to the plots in the main text. Scripts of simulations underlying this figure can be found at doi 10.17863/CAM.120831.
(TIF)

**S5 Fig. Branch counts of *SIP1* lines.** (A) Photograph of a representative plant from the genotypes indicated, taken at terminal flowering. **(B)** Primary rosette and **(C)** total (cauline and rosette) branch number at terminal flowering in plants of the genotypes indicated grown under long day conditions, $n = 53$–68. The boxes span the first to third quartile, the line represents the median. The whiskers indicate the variability outside the upper and lower quartiles. A generalized linear multilevel model was used to assess the effect of genotype while accounting for variability across experiments. Comparisons were adjusted using Bonferroni correction. Different letters indicate statistically significant differences at $p < 0.05$. Data and analysis underlying this figure can be found at doi 10.17863/CAM.120831.
(TIF)

**S6 Fig. Replicate of qPCR from** Fig 7E**: relative expression of *BRC1* as measured by qRT-PCR in Col-0, *SIP1pin1*, *smxl678* and *SIP1pin1smxl678* in small buds (>2.5 mm, presumed inactive) and big buds (<5 mm, presumed active).** Buds were harvested from cauline nodes of whole Arabidopsis plants. Each point represents a biological replicate, with the superimposed mean and associated standard error. Each biological replicate corresponds to the expression level measured in at least 10 pooled buds. Statistical comparisons were made on the log transformed data, using two-way ANOVA with Bonferroni correction. Different letters indicate statistically significant differences at $p < 0.05$. Data and analysis underlying this figure can be found at doi 10.17863/CAM.120831.
(TIF)

**S7 Fig. Bulk auxin transport and membrane fluorescence of *SIP1pin1pin7* and *SIP1pin1brc1brc2*.** (A) Bulk auxin transport through 15 mm basal inflorescence stem segments of 6–8 week old plants of the genotypes indicated. Transport was determined as accumulation of 3H auxin in the basal 5 mm of stem after 18 h of incubation of the apical end 22 nM 3H-IAA, $n = 57$–68. **(B)** Boxplot of the mean fluorescence intensity of basal plasma membranes of xylem parenchyma cells from at least 8 stems per genotype. For each stem, the mean fluorescence intensity represents the mean of the 5 brightest membranes for that stem. Fluorescence intensity was extracted using the ZEN imaging software, $n = 9$–37 stems. Letters indicate statistically significant differences at $p < 0.05$ from a multi-level model with Bonferroni correction. Data and analysis underlying this figure can be found at doi 10.17863/CAM.120831.
(TIF)

**S8 Fig. Strigolactone responsiveness of *SIP1pin1pin7* and *SIP1pin1brc1brc2*.** (A) Mitchison plot of 2-node explants from two-three experimental replicates of Col-0, *SIP1pin1, SIP1pin1pin7, brc1brc2, SIP1pinbrc1brc2,* and *d14*, with basal mock or 5 μM GR24 treatment. Buds were measured daily for 12 or 13 days, and the final length of buds in each explant is shown with a black dot, $n = 41$–80. **(B)** Relative growth index (RGI) for the data in panel **A**. The black dot indicates the mean. The RGI is calculated as the length of the longest branch divided by the summed length of both branches. Only active explants are used to calculate the RGI. Active explants are those with at least one active bud (a bud is considered active if it has a maximum growth rate $> 2.5$ mm/day), $n = 12$–80. **(C)** Percentage of explants with at least one active bud in each of the three experimental replicates. **(D)** Maximum growth rate and **(E)** Number of lag days for the longer buds in panel **A**. The number of lag days is calculated as the number of days before buds reach a growth rate of 2.5 mm/day. The data are plotted for active buds only, $n = 12$–80. In panels **B–E**, *S1p1p7* and *S1p1b1b2* are used as shorthands for *SIP1pin1pin7*, and *SIP1pin1brc1brc2*, respectively. Asterisks indicate statistically significant differences at $p < 0.05$ from a multi-level model with custom contrast on the comparisons of interest and Bonferroni corrections. Data and analysis underlying this figure can be found at doi 10.17863/CAM.120831.
(TIF)

**S1 Appendix. Mathematical appendix detailing the model derivation.**
(PDF)

## Acknowledgments

We thank Chris Whitewoods for critical reading of the manuscript, as well as Geneviève Hines and Graeme Mitchison for valuable discussions underpinning the project.

## Author contributions

**Conceptualization:** Zoe Nahas, Katie Abley, James C. W. Locke, Ottoline Leyser.

**Data curation:** Zoe Nahas.

**Formal analysis:** Zoe Nahas.

**Funding acquisition:** James C. W. Locke, Ottoline Leyser.

**Investigation:** Zoe Nahas, Anthony John Bridgen, Dora L. Cano-Ramirez, Madeleine Seale.

**Methodology:** Zoe Nahas, Torkel E. Loman, Katie Abley.

**Resources:** Anthony John Bridgen, Torkel E. Loman, Jean Dillon, Fabrizio Ticchiarelli.

**Software:** Zoe Nahas, Torkel E. Loman.

**Supervision:** James C. W. Locke, Ottoline Leyser.

**Validation:** Zoe Nahas, Anthony John Bridgen.

**Visualization:** Zoe Nahas.

**Writing – original draft:** Zoe Nahas, James C. W. Locke, Ottoline Leyser.

**Writing – review & editing:** Zoe Nahas, Anthony John Bridgen, Torkel E. Loman, Jean Dillon, Madeleine Seale, James C. W. Locke, Ottoline Leyser.

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
