## [Editor Report · Decision Letter 0]

22 Apr 2025

Dear Dr Nahas,

Thank you for submitting your manuscript entitled "A BRC1-modulated switch in auxin transport accounts for the competition between Arabidopsis axillary buds" for consideration as a Research Article by PLOS Biology.

Your manuscript has now been evaluated by the PLOS Biology editorial staff as well as by an academic editor with relevant expertise and I am writing to let you know that we would like to send your submission out for external peer review.

Once your full submission is complete, your paper will undergo a series of checks in preparation for peer review. After your manuscript has passed the checks it will be sent out for review. To provide the metadata for your submission, please Login to Editorial Manager (https://www.editorialmanager.com/pbiology) within two working days, i.e. by Apr 24 2025 11:59PM.

Kind regards,

Ines

--

Ines Alvarez-Garcia, PhD

Senior Editor

PLOS Biology

---

## [Decision Letter · Decision Letter 1]

13 Jun 2025

Dear Dr Nahas,

Thank you for your patience while your manuscript entitled "A BRC1-modulated switch in auxin transport accounts for the competition between Arabidopsis axillary buds" went through peer-review at PLOS Biology. Your manuscript has now been evaluated by the PLOS Biology editors, an Academic Editor with relevant expertise, and by two independent reviewers.

The reviews are attached below. As you will see, the reviewers find the results interesting, but they also raise several points that would need to be addressed before we can consider the manuscript for publication. Reviewer 1 mentions that the interpretation of some of the results is confusing and that BRC1 modulates auxin transport is not directly demonstrated. The reviewer also offers an alternative explanation by which BRC1 impacts auxin availability rather than transport itself, and has some doubts regarding the role of BRC2, which is not analysed. Reviewer 2 raises three points that need clarification: s/he notes that in the model there is no parameter that is dependent on the amount of auxin in the system and wonders if sugar could be playing a role; whether or not the effect could be due to boosting the total level of auxin in the system; and if both buds could reach the ‘tipping point’ of canalisation faster, it would be more auxin available. In addition, the reviewer makes suggestions to improve readability and the proposed model.

In light of the reviews, we are pleased to offer you the opportunity to address the comments from the reviewers in a revision that we anticipate should not take you very long. We will then assess your revised manuscript and your response to the reviewers' comments with our Academic Editor aiming to avoid further rounds of peer-review, although we might need to consult with the reviewers, depending on the nature of the revisions.

**IMPORTANT - SUBMITTING YOUR REVISION**

3. Resubmission Checklist

a) *PLOS Data Policy*

b) *Published Peer Review*

d) *Blurb*

Please also provide a blurb which (if accepted) will be included in our weekly and monthly Electronic Table of Contents, sent out to readers of PLOS Biology, and may be used to promote your article in social media. The blurb should be about 30-40 words long and is subject to editorial changes. It should, without exaggeration, entice people to read your manuscript. It should not be redundant with the title and should not contain acronyms or abbreviations. For examples, view our author guidelines: https://journals.plos.org/plosbiology/s/revising-your-manuscript#loc-blurb

Sincerely,

Ines

--

Ines Alvarez-Garcia, PhD

Senior Editor

PLOS Biology

Reviewers' comments

Rev. 1:

This manuscript studies how auxin transport and the transcription factor BRC1 interact to regulate axillary bud outgrowth. Both responses are controlled by strigolactone (SL) signaling. Using two-node explant bud competition experiments in several genetic backgrounds -brc1brc2, smxl678, and the brc1brc2 smxl678- and in response to GR24 (synthetic SL) treatments the authors investigate in depth how buds compete and respond, and how SL, BRC1 and the auxin. Transporter PIN1 participate in this process. These experiments allow them to generate and validate, also in detail, a mathematical model aimed at understanding the molecular components influencing the lag phase and the rapid outgrowth phase of bud activation. The extensive data, together with previous work from the same group, shows that SL acts through both a BRC1/2-dependent and a BRC1/2-independent pathway during bud outgrowth. In particular, during the rapid outgrowth phase, SL appears to influence auxin transport in a BRC1-independent manner.

The authors also design an elegant experiment, in which a domain of PIN1 is replaced with the corresponding domain of PIN3 to generate a modified PIN1 less responsive to SL (SIP1). The authors show that SIP1 protein is indeed more resistant to SL. However, interpretation of these results becomes confusing due to a partial SL sensitivity of the protein, noticeable in the response of buds to SL, and to impaired auxin transport in the SIP1 lines. This complicates the interpretation of all the experiments performed with this line (eg. Figure 8, Figures S6, S9).

One of the main conclusions of the paper, also featured in the title of the manuscript, is that BRC1 modulates auxin transport. However, this point is not directly demonstrated. It is derived from interpretations of the modeling outputs and the behavior of buds in the competition assays. In the discussion, the authors note this limitation and acknowledge that no auxin transporters are currently known to be transcriptional targets of BRC1. They suggest a possible post-translational mechanism, though this remains speculative. Besides, one known BRC1 target, GH3.5, reduces active IAA levels through conjugation, which could offer an alternative explanation (not considered) by which BRC1 impacts auxin availability rather than transport per se.

It is also worth noting that while all genetic experiments are carried out using the brc1 brc2 double mutant, the qPCR analyses and the modeling (and the title) focus on BRC1. This raises the question of why is BRC2 excluded from these parts of the study.

Rev. 2:

The manuscript by Nahas et al is an elegant integration of computational modeling and experimental validation—both aimed at formalizing a minimal regulatory model for competition between buds. The authors have done a significant amount of work, and it is all thoughtfully designed, appropriately controlled and interpreted with reasonable cautions. The engineering of a largely SL-independent PIN1 was excellent, although more work could be done to enhance interpretability, as the authors describe. Available mutants are used to great effect throughout.

The utility of a model with minimal parameters cannot be overstated, especially for hypothesis generation, and the authors have done an admirable job of capturing decades of findings within a strikingly simple architecture. Essentially, the authors argue that both model and validating experiments point to two significant tuning knobs that together determine whether one or both buds will grow in a 2-node explant: (1) how much auxin efflux transporter is removed from the membrane, and (2) what is the basal rate of auxin efflux. They go on to show that strigolactones primarily control the former, while BRC1 is largely acting on the latter.

I have no major concerns, but a few questions:

1) If I am reading the model correctly, there is no parameter that is dependent on the amount of auxin in the system. If I understand that correctly, I wonder if that partially explains the unexpected behavior of the smxl678 mutant in the 2 node vs 1 node assay. The authors argue that sugar could be playing a role here, and I wonder if that effect could be by boosting the total level of auxin in the system. If there was more overall auxin, perhaps both buds could reach the 'tipping point' of canalization more quickly.

2) It is a personal preference, but in the interest of readability, I would love to see fewer abbreviations used throughout the text. Maximum growth rate is somewhat cumbersome to spell out, but MGR is not really an improvement in my opinion. Similarly, in Figures like S1, it would be wonderful to have the meanings of each parameter being tested spelled out in the titles of each graph or at least in the figure legend to avoid having to flip back and forth.

3) It would be useful to have the authors speculate a bit on how the current model might succeed or struggle in a more complex system of higher numbers of buds, and other considerations that might need to be incorporated to apply the model to other plants (e.g., different shoot architectures).

---

## [Editor Report · Decision Letter 2]

7 Aug 2025

Dear Dr Nahas,

Thank you for your patience while we considered your revised manuscript entitled "A BRC1-modulated switch in auxin efflux accounts for the competition between Arabidopsis axillary buds" for publication as a Research Article at PLOS Biology. This revised version of your manuscript has been evaluated by the PLOS Biology editors and by the Academic Editor.

Based on our Academic Editor's assessment of your revision, we are likely to accept this manuscript for publication, provided you satisfactorily address the data and other policy-related requests stated below my signature.

We expect to receive your revised manuscript within two weeks.

*Published Peer Review History*

*Press*

Sincerely,

Ines

--

Ines Alvarez-Garcia, PhD

Senior Editor

PLOS Biology

Fig. 2B-E; Fig. 3C; Fig. 4A; Fig. 5B, C; Fig. 6B, C; Fig. 7A-E; Fig. 8C, E; Fig. 9C, F; Fig. S5B, C; Fig. S6; Fig. S7A, B and Fig. S8A-E

CODE POLICY

Please note that we do need a DOI number for the code, thus we suggest you deposit it also in Zenodo, which will generate a DOI number and you will need to provide in the Data Accessibility Statement

---

## [Editor Report · Decision Letter 3]

1 Sep 2025

Dear Dr Nahas,

Thank you for the submission of your revised Research Article entitled "A BRC1-modulated switch in auxin efflux accounts for the competition between Arabidopsis axillary buds" for publication in PLOS Biology. On behalf of my colleagues and the Academic Editor, Xinnian Dong, I am delighted to let you know that we can in principle accept your manuscript for publication, provided you address any remaining formatting and reporting issues. These will be detailed in an email you should receive within 2-3 business days from our colleagues in the journal operations team; no action is required from you until then. Please note that we will not be able to formally accept your manuscript and schedule it for publication until you have completed any requested changes.

PRESS

Sincerely, 

Ines

--

Ines Alvarez-Garcia, PhD

Senior Editor

PLOS Biology
